# The invariant cleavage pattern displayed by ascidian embryos depends on spindle positioning along the cell's longest axis in the apical plane and relies on asynchronous cell divisions

**Rémi Dumollard[1]\*, Nicolas Minc[2], Gregory Salez[1], Sameh Ben Aicha[1], Faisal Bekkouche[1], Céline Hebras[1], Lydia Besnardeau[1], Alex McDougall[1]\***

[1]Laboratoire de Biologie du Développement de Villefranche-sur-mer (LBDV) UMR7009, Sorbonne Universités, Université Pierre-et-Marie-Curie, CNRS, Villefranche sur mer, France; [2]Institut Jacques Monod, UMR7592 CNRS, Paris, France

**Abstract** The ascidian embryo is an ideal system to investigate how cell position is determined during embryogenesis. Using 3D timelapse imaging and computational methods we analyzed the planar cell divisions in ascidian early embryos and found that spindles in every cell tend to align at metaphase in the long length of the apical surface except in cells undergoing unequal cleavage. Furthermore, the invariant and conserved cleavage pattern of ascidian embryos was found to consist in alternate planar cell divisions between ectoderm and endomesoderm. In order to test the importance of alternate cell divisions we manipulated zygotic transcription induced by $\beta$-catenin or downregulated wee1 activity, both of which abolish this cell cycle asynchrony. Crucially, abolishing cell cycle asynchrony consistently disrupted the spindle orienting mechanism underpinning the invariant cleavage pattern. Our results demonstrate how an evolutionary conserved cell cycle asynchrony maintains the invariant cleavage pattern driving morphogenesis of the ascidian blastula.

**\*For correspondence:** remi. dumollard@obs-vlfr.fr (RD); dougall@obs-vlfr.fr (AM)

**Competing interests:** The authors declare that no competing interests exist.

## Introduction

In chordate embryos the functional pattern of cells is generated before gastrulation such that a fate map for all chordate embryos at the blastula stage predicts that cells in different positions will give rise to new cell types and layers that are important for morphogenesis (*Kourakis and Smith, 2005*). Invertebrate chordate embryos of the ascidian display a similar fate map to other chordates even though their blastulae are composed of only 64 cells rather than several thousand cells typical of other chordates (*Kourakis and Smith, 2005*; *Lemaire et al., 2008*). Due to its conserved fate map yet small cell number, the ascidian embryo is an ideal system to elucidate mechanisms underpinning cell positioning during morphogenesis of a chordate blastula. Because ascidian embryos display an invariant cleavage pattern with no cell migration or cell death up to the time of gastrulation, cell division plane orientation is important for pattern formation (*McDougall et al., 2015*). In addition, although the invariant-cleavage pattern displayed by asicidian embryos is specific to ascidians, the lophotrochozoa also display stereotyped spiral cleavage patterns that may employ similar rules as the ones we address here in the ascidian (*Rabinowitz and Lambert, 2010*; *Brun-Usan et al., 2017*). All blastomeres are fate restricted in the 64 cell ascidian blastula and most cell divisions at the 32 cell stage are fate segregating asymmetric divisions (*Lemaire et al., 2008*; *Lemaire, 2009*). Such

**eLife digest** The position of cells within an embryo early in development determines what type of cells they will become in the fully formed embryo. The embryos of ascidians, commonly known as sea squirts, are ideal for studying what influences cell positioning. These embryos consist of a small number of cells that divide according to an "invariant cleavage pattern", which means that the positioning and timing of the cell divisions is identical between different individuals of the same species. The pattern of cell division is also largely the same across different ascidian species, which raises questions about how it is controlled.

When a cell divides, a structure called the spindle forms inside it to distribute copies of the cell's genetic material between the new cells. The orientation of the spindle determines the direction in which the cell will divide. Now, by combining 3D imaging of living ascidian embryos with computational modeling, Dumollard et al. show that the spindles in every equally dividing cell tend to all align in the long length of the cell's "apical" surface. Such alignment allows the cells to be on the outside of the embryo and implements the ascidian invariant cleavage pattern.

The cells in the embryo do not all divide at the same time. Indeed, the shape of the cells (and especially their apical surface) depends on two layers of cells in the embryo not dividing at the same time; instead, periods of cell division alternate between the layers. A network of genes in the embryo regulates the timing of these cell divisions and makes it possible for the cells to divide according to an invariant cleavage pattern.

However, this network of genes is not the only control mechanism that shapes the early embryo. A structure found in egg cells (and hence produced by the embryo's mother) causes cells at the rear of the embryo to divide unequally, and this influences the shape of all the cells in the embryo. Thus it appears that maternal mechanisms work alongside the embryo's gene network to shape the early embryo.

The next step will be to determine how physical forces – for example, from the cells pressing against each other – influence the position of the embryo's cells. How do gene networks relay the biomechanical properties of the embryo to help it take shape?

fate-segregating asymmetric cell divisions rely on precisely regulated cell divisions partitioning maternal determinants (muscle lineage) or allowing local cell-cell contacts for polarised inductive cell fate specification e.g. neural lineage induction (*Kumano and Nishida, 2007*; *Lemaire, 2009*).

The invariant cleavage pattern of ascidian embryos is a relatively simple morphogenetic process operating at the level of the whole embryo that is amenable to genetic analysis in order to identify the gene regulatory networks (GRNs) controlling cell division orientation. The overall invariant temporal and spatial pattern of cell divisions in ascidians is even conserved between distantly-related species. Ascidians are split into three orders that diverged more than 350 millions years ago (aplousobranch, phlebobranch and stolidobranch), and it has been estimated that non-coding DNA sequences from two distinct ascidian species can be as different from each other as fish are from mammals (*Stolfi et al., 2014*). Distantly-related species of ascidian also show the same relative cell cycle asynchrony since the 24 cell stage embryo is common to both phlebobranch (*Ciona/Phallusia*) and stolidobranch (*Halocynthia/Styela*) ascidians. In *Phallusia* this cell cycle asynchrony is induced by a GRN dependent upon nuclear accumulation of $\beta$-catenin in six vegetal cells of the 16 cell stage embryo (*Dumollard et al., 2013*). How such stereotyped cell cycle asynchrony has been conserved in distantly-related ascidians is presently unknown, but it is interesting to note that $\beta$-catenin becomes nuclear in vegetal blastomeres in both *Ciona* and *Halocynthia* embryos at the 16 cell stage (*Kawai et al., 2007*; *Hudson et al., 2013*).

Mitotic spindles align relative to a number of cues that display a competitive hierarchal relationship with one another. For example, an underlying mechanism known as the long-axis rule based upon microtubule behavior and motors (reviewed in *Minc and Piel, 2012*) causes animal cell to divide orthogonal to their long axis as was noted more than a century ago by Hertwig (*Hertwig, 1893*). This geometric long-axis rule can be altered by cortical polarity cues such as lateral junctions (*Nakajima et al., 2013*; *Ragkousi and Gibson, 2014*) or the apical cortex in asymmetrically

dividing *Drosophila* neuroblasts (*Siller and Doe, 2009*). During planar cell divisions in epithelia and endothelia, a lateral belt of LGN/NuMA coupled with the exclusion of LGN/NuMA from the apical cortex causes planar spindle orientation (*Zheng et al., 2010*; *Morin and Bellaïche, 2011*). After acquiring a planar orientation the spindle rotates in the apical plane to find its final position at metaphase. Spindle orientation in the apical plane will set cell position in the epithelium and is regulated by apical cell shape (*Ragkousi and Gibson, 2014*). Because of mitotic cell rounding in cultured cells and some epithelia, apical cell shape at metaphase may become completely round (*Lancaster and Baum, 2014*). In these cells, the spindle aligns with the long axis of the cell during interphase which is memorized during mitotic cell rounding via retraction fibers in cultured cells (*Théry and Bornens, 2008*) or LGN/NuMA-rich tricellular junctions in *Drosophila* epithelia (*Bosveld et al., 2016*). Alternatively, mitotic cell rounding is less pronounced in the squamous epithelia such as the enveloping cell layer (EVL) of Zebrafish gastrulae which maintain a long axis at metaphase to orient the mitotic spindle (*Campinho et al., 2013*). Mitotic cell rounding does not seem to occur in the *Xenopus* blastula (*Strauss et al., 2006*) and remains poorly documented in blastulae of other species (*Xiong et al., 2014*). A computational approach revealed very recently that the first 4 cell divisions in ascidian embryos may follow a geometric rule in a similar manner to early *Xenopus*, Zebrafish or sea urchin embryos (*Pierre et al., 2016*). Major cell shape changes have been noted during the 32–44 cell stage in ascidian (*Ciona*) embryos (*Tassy et al., 2006*) suggesting that mitotic cell rounding may occur in cells of the ascidian blastula. However, the impact of cell cycle asynchrony or mitotic cell rounding on mitotic spindle orientation in cells of the ascidian blastula have not been studied so far.

Ascidian embryos display an invariant cleavage pattern up to the 64 cell stage such that both the orientation of cell division and the relative timing of cell division in the different lineages are predictable. In order to determine if mitotic spindles aligned with the cells longest length in the apical plane we extracted the apical plane of every blastomere and assessed with a 2D computational model whether geometry of the apical surface can regulate spindle orientation in the apical plane to set cell positioning. Finally, since both phlebobranch and stolidobranch ascidians display a 24 cell stage indicating that cell cycle asynchrony begins at the 16 cell stage, we assessed what impact reducing this cell cycle asynchrony has on spindle orientation and the invariant cleavage pattern. We perturbed the GRN driving this cell cycle asynchrony to create quasi-synchronous cell cycles and determined the impact on spindle orientation in the apical plane. In this study we provide evidence that spindles align parallel to the apical surface and along the longest length of the apical surface of the blastomeres. We also show that the invariant cleavage pattern is disrupted when the asynchronous cell cycles are made more synchronous.

## Results

### Spindle rotations during oriented cell divisions underlie the invariant cleavage pattern of ascidian embryos

Ascidian embryos undergo a very stereotyped development consisting of unequal cleavages and symmetric cell divisions (*Lemaire, 2009*; *McDougall et al., 2011*). At the 64 cell stage all cells of the ascidian blastula face the outside of the embryo (*Figure 1A*) suggesting that all cell divisions are parallel to the apical surface and that embryonic cleavage proceeds through planar cell divisions to generate the ascidian blastula. In unstretched epithelia (and in early embryos) daughter cells divide orthogonally to the mitotic orientation of their mother cell reflected in one cell generating a square of 4 cells upon two planar cell divisions (*Wyatt et al., 2015*). Cell division is said to be oriented (called oriented cell division: OCD) when daughter cells divide in the same orientation as their mother (Strome, 1993; *Wyatt et al., 2015*). By analysing the pattern of 4 grand-daughter cells generated by two successive cell divisions from the 16 cell stage in *Phallusia* embryos (see *McDougall et al., 2015* for details) we found that some blastomeres do not divide orthogonally to their mother. *Figure 1A* shows a virtual *Phallusia mammillata* embryo (*Tassy et al., 2006*) with color-coded lineages. When following two successive cell divisions from the 16 cell stage it can be observed at the 64 cell stage that some groups of 4 grand-daughters form a square pattern (lineages b5.3, b5.4 and A5.2 shown in blue and pink, *Figure 1A*) suggesting that two cell divisions orthogonal to each other occurred. In contrast, the grand-daughters of B5.1, B5.2 and a5.3 (brown) form a T pattern suggesting that the spindle of one of the two daughter cells is in the same

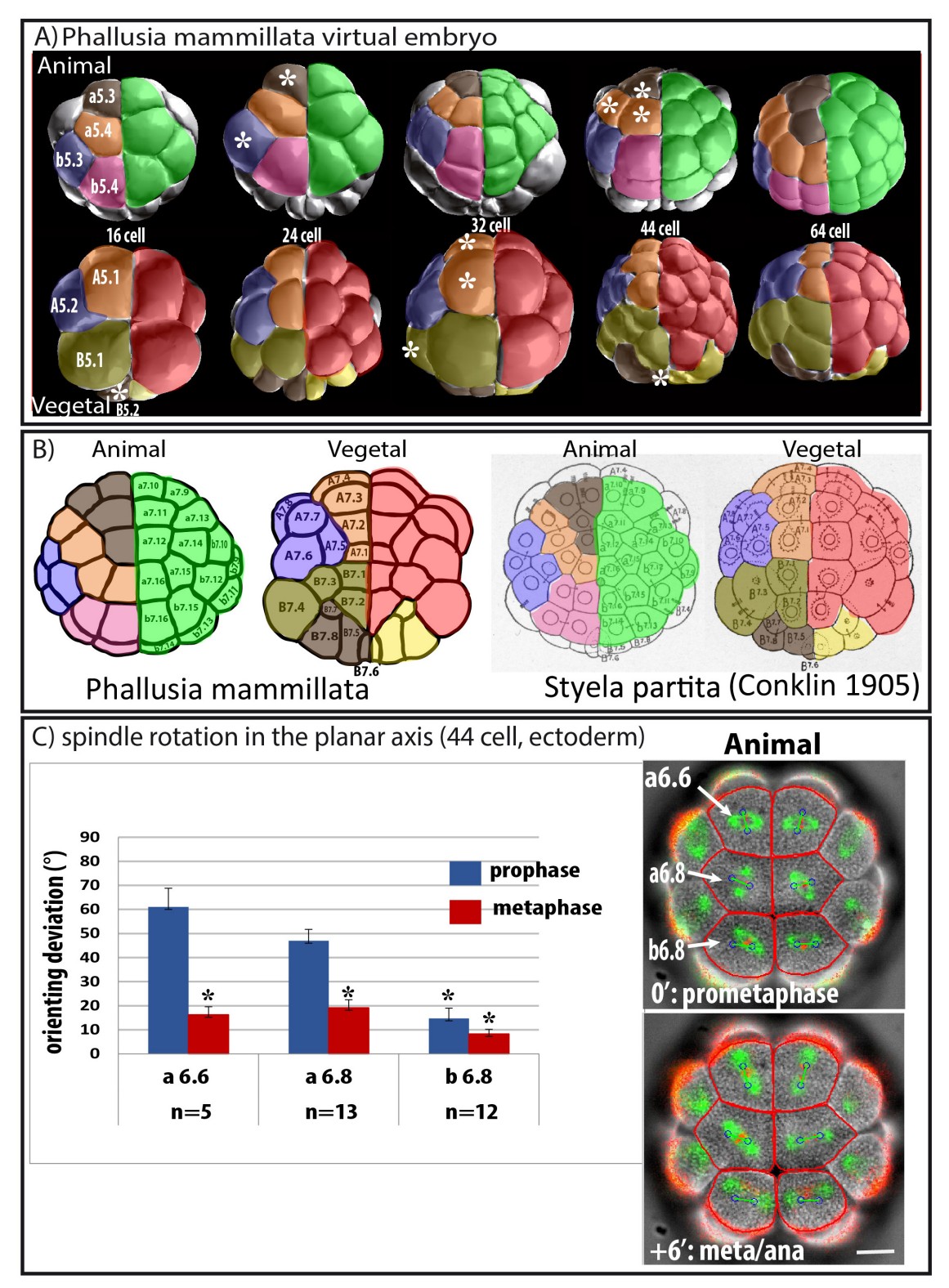

**Figure 1.** Predicted oriented cell divisions (OCD) in ascidian embryos. (**A**) Images taken from virtual *Phallusia mammillata* embryos (obtained from http://www.aniseed.cnrs.fr/aniseed/download/download_3dve) showing the different embryonic stages. Top row: Animal hemisphere, bottom row: Vegetal hemisphere. The right side of embryos is color coded for germ layers at the 16 cell stage: Ectoderm is in green, endomesoderm in red and germ lineage in yellow. The left side of embryo is color coded according to type of lineages. Lineages displaying square arrangements of 4 cells at 64

*Figure 1 continued on next page*

*Figure 1 continued*

cell stage are shown in blue (b5.3, A5.2) and pink (b5.4). Lineages displaying T arrangements are depicted in light (B5.1) and dark (a5.3, B5.2) brown. Lineages displaying linear arrangements of cell are depicted in orange (a5.4, A5.1). (B) Schematic drawing showing 64 cell stage embryos of *Phallusia mammillata* (left, outlines from an embryo stained with Cell Mask Orange) and of *Styela partita* (*Conklin, 1905*). The names of each blastomere are depicted to show conservation of cell positions between the two distant ascidian species. Same color coding as in A. (C) Spindle rotation in the ectoderm (Animal hemisphere) at the 44 cell stage. Time lapse epifluorescence imaging of a *P. mammillata* embryo expressing MAP7::GFP to monitor mitotic spindles and H2B::mRFP1 to monitor DNA (superimposed on the BF image). In red are the cell's outline drawn using the BF image during the running of the computational model. Blue circles joined by a green bar represent mitotic spindles predicted by the computational model. Scale bar = 20 µm. Bar graph showing quantification of the angle difference between observed and predicted spindles (orienting deviation). Black asterisks denote statistical difference with the value for a6.6 at prophase/prometaphase (student test; *p<0.05; ***p<0.0001). n represents the number of blastomeres analysed with the computational model.

orientation as the spindle of its mother (indicating OCD in this cell). Finally the grand-daughters of A5.1 and a5.4 (orange) form a line indicating that two OCDs occurred in these lineages (*Figure 1A and B*, see also *McDougall et al., 2015*). Using this strategy we could identify three cells undergoing OCD (asterisks in *Figure 1A*) at the 16–24 cell stage (a5.3; a5.4; B5.2) and seven cells at the 32–44 cell stage (A6.1; A6.2; a6.6; a6.7; a6.8; B6.2 and B6.3). Strikingly the square, T and linear patterns observed at the 64 cell stage are perfectly conserved in *Styela partita* (*Figure 1B*) and *Ciona intestinalis* (*McDougall et al., 2015*), showing that the pattern of planar cell divisions in early ascidian embryos may be perfectly conserved.

Time lapse imaging of mitotic spindles in live ascidian embryos revealed that spindle rotation accompanies unequal cleavages in the germ lineage (B5.2; B63; *Prodon et al., 2010*) but also in several other lineages where we predicted OCDs (for a5.3; b5.3; a6.6; a6.7; a6.8 see *Video 1* and *Figure 1C* and for A6.1; A6.2 see *Negishi and Yasuo, 2015*). In the experiments depicted in *Figures 1C* and *2D* epifluorescence imaging is used and only the plane of imaging is analysed. The apical surface of a6.6, a6.8 and b6.8 cells can be imaged by our 2D imaging protocol as both spindle poles remain in the plane of imaging during the whole of mitosis. We consistently observed spindle rotation in the apical plane of a6.6 and a6.8 which are predicted to undergo OCD (*Figure 1C*, *Video 1*). Spindle rotation in A6.1, A.62 (*Negishi and Yasuo, 2015*) and a6.7 (data not shown) which also display OCD is also limited to the apical plane of the cell. In contrast, blastomeres not displaying OCD such as b6.8 blastomeres show no major spindle rotation (*Figure 1C*, *Video 1*).

To test whether spindle rotation is influenced by the geometry of the apical surface of the cell, we used a 2D computational model that predicts the preferred spindle orientation based on cell shape (*Minc et al., 2011*; *Minc and Piel, 2012*; *Campinho et al., 2013*; *Bosveld et al., 2016*). Using the outline of cells (in red), the computational model outputs a predicted spindle position and orientation (blue circles joined by a green line) which do not change between prophase and anaphase (*Figure 1C*). By comparing the angle between the observed and predicted spindle orientations (orienting deviation) it can be observed that at prophase/prometaphase (i. e., before spindle rotation), apical cell geometry poorly predicts spindle orientation in a6.6 and

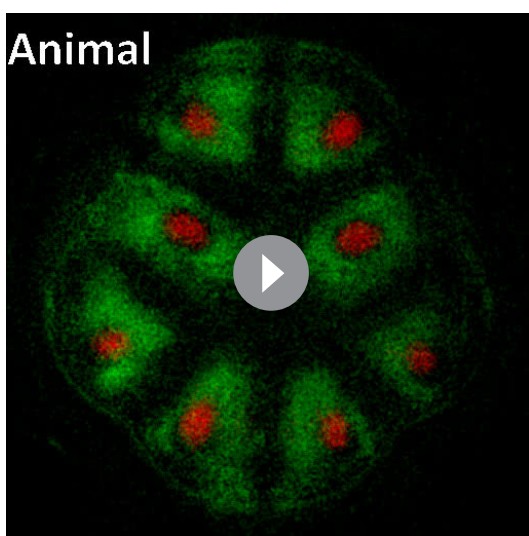

**Video 1.** Spindle rotation in the ectoderm in cells undergoing oriented cell divisions. Movie showing live imaging of a *Phallusia mammillata* embryo injected with RNAs coding for MAP7::GFP (in green) and H2B:: mRfp1 (in red). (z-stacks taken 2 min apart). View of the ectoderm showing mitoses of 24–32 cell stage and 44–64 cell stage. Spindle rotation is clearly visible in six blastomeres at mitosis 44–64 cell stage (a6.6; a6.7; a6.8).

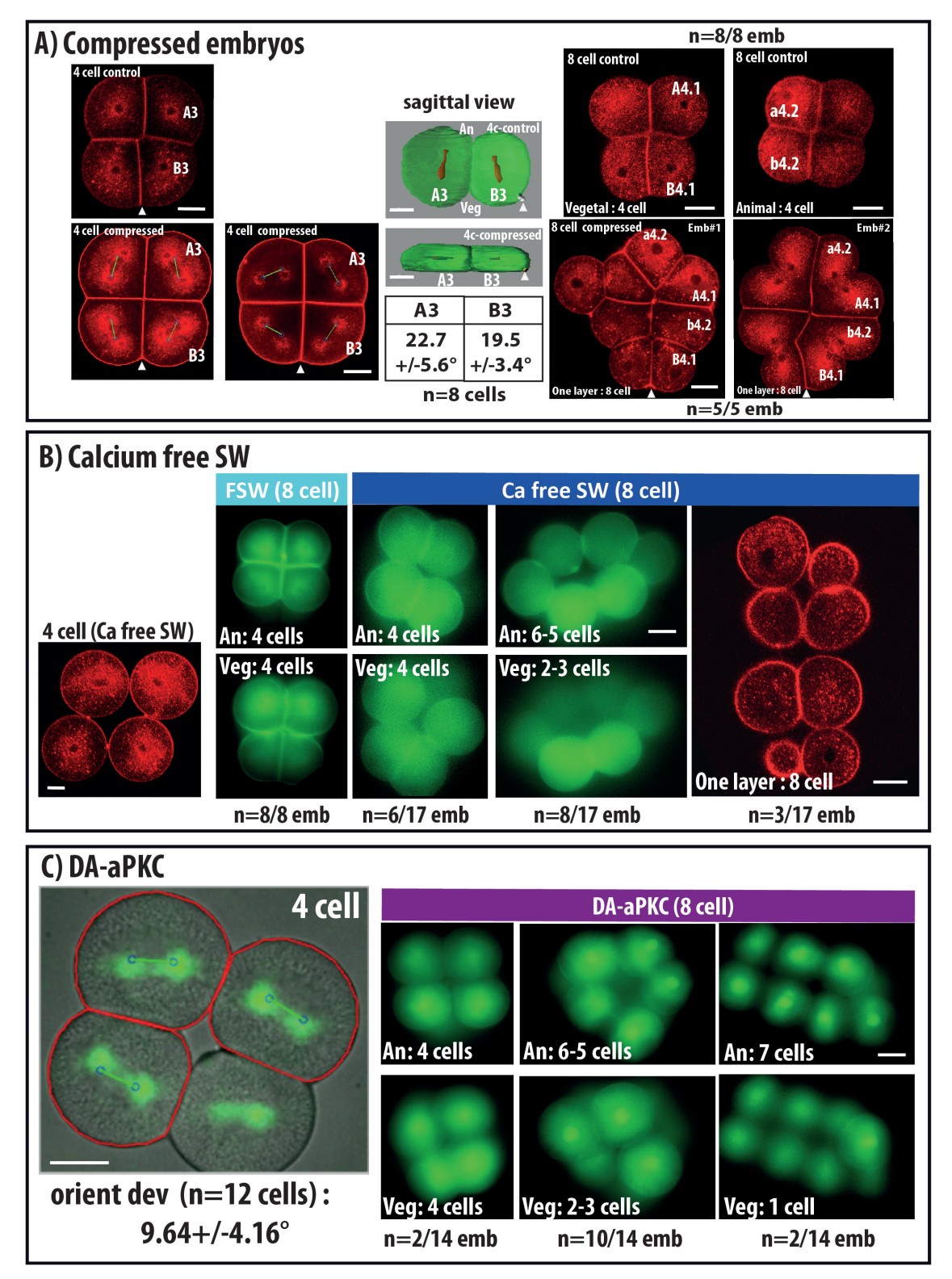

**Figure 2.** Changing cell shape in the embryo by compressing embryos or removing cell adhesion. (**A**) Compressing embryos: Left: CellMask images of 4 cell-stage embryos in control (4 cell control) or compressed (4 cell compress) conditions. A3, B3 are names of blastomeres, arrowheads indicate the position of the CAB (marking the posterior pole of the embryo). In the images of 4 cell compressed embryos, predicted spindles are shown (blue circles joined by a green line). Scale bar = 30 µm. Middle: sagittal views of control and compressed embryos after 3D rendering on Imaris. Plasma membrane

*Figure 2 continued on next page*

*Figure 2 continued*

is in green and spindles in red, arrowheads indicate the position of the CAB. Table shows orienting deviation measured in A3 and B3 blastomeres in compressed embryos (n = 8 cells taken from five embryos). Right: Top: CellMask images of control embryo at the 8 cell stage with a Vegetal layer of 4 cells and an Animal layer of 4 cells (same embryo shown). CellMask images of 2 different compressed embryos at the 8 cell stage showing one layer of 8 cells. Arrowheads show the position of the CAB. Scale bar = 30 μm. (**B**) Culture in Ca2+ free sea water to remove cell adhesion: CellMask image of 4 cell stage embryo cultured in Ca2+ free sea water from the one cell stage. Like in a control embryo, the 4 cells are arranged in one plane. FSW: 8 cell stage embryo cultured in filtered sea water (FSW) exhibiting 2 layers of 4 animal and four vegetal cells (n = 8/8 embryos). Ca-Free SW: 8 cell stage embryos cultured in Ca2+-free sea water from the 2 cell stage exhibiting variable morphologies comprising either wild type morphology (4 animal and four vegetal cells, n = 6 out of 17 embryos) or affected morphologies with 6–5 animal and 2–3 vegetal cells (n = 8 out of 17 embryos) or even one layer of 8 cells (3 out of 17 embryos). Scale bars = 30 μm. (**C**) DA-aPKC: Image of a 4 cell stage embryo showing embryo morphology (BF image) and spindle positions (imaged with Venus::Tpx2) as well as predicted spindle positions superimposed (blue circles joined by a green line: for those cells where both spindle poles were in the imaging plane). Scale bar = 30 μm. 8 cell stage embryos expressing DA-aPKC::Venus which exhibit variable morphologies comprising either wild type morphology (4 animal and four vegetal cells, n = 2 out of 14 embryos) or affected morphologies with 6–5 animal and 2–3 vegetal cells (n = 10 out of 14 embryos) or 7 and 1 cells (2 out of 14 embryos). Scale bar = 30 μm.

a6.8 (orienting deviation: 61.0° in a6.6 and 47.0° in a6.8) whereas apical cell geometry predicts reliably spindle position in b6.8 (orienting deviation: 14.7°). In contrast, at metaphase, all observed spindles show an orienting deviation of less than 20° indicating that apical cell shape predicts reliably spindle orientation in the apical plane (*Figure 1C*). These observations suggest that mitotic spindles in a6.6 and a6.8 are not aligned with the long length of the cell in the apical plane at prophase and they rotate during mitosis to align with the long length of the cell's apical surface at metaphase. In contrast, in b6.8 (not displaying OCD) the spindle is aligned in the long length of the cell's apical surface already in prophase.

## Cell shape and cell adhesion support the invariant cleavage pattern at mitosis 4–8 cell

In order to confirm that cell shape can influence spindle positioning in early ascidian embryos we changed blastomeres shapes by compressing embryos or by inhibiting cell adhesion (*Figure 2*). Mitosis from 4 to 8 cells transforms a single layered 4 cell stage embryo into an 8 cell embryo made of 2 layers of cells comprising 4 animal and four vegetal cells as all mitotic spindles align along the Animal-Vegetal axis of the embryo (*Figure 2*, *Negishi et al., 2007*; *Pierre et al., 2016*). Compressing the 4 cell stage embryo along the Animal-Vegetal axis creates a flattened embryo (*Figure 2A*, '4-cell compressed') which, upon cell division, gives rise to a single layer of 8 cells. Computational analysis shows a good prediction of spindle positioning by cell shape in these embryos (orienting deviation: 22.7 ± 5.6° for A3 blastomere and 19.5 ± 3.4° for B3) (*Figure 2A*). Therefore mitotic spindle orientation in these compressed embryos tends to follow the newly created long axis of the cell.

We then removed cell adhesion between blastomeres by culturing embryos in Ca2+-free sea water from the one-cell stage (*Figure 2B*) or by inhibiting basolateral membrane formation using a dominant active form of aPKC (*Sabherwal et al., 2009*, *Figure 2C*). Both treatments reduced drastically cell adhesion from the 2 cell stage resulting in rounder blastomeres. Spindle orientation did not seem affected during the mitosis from 2 to 4 cell stage (data not shown) and 4 cell stage embryos with reduced cell adhesion show no blastomere positional changes as they are still made by a single layer of 4 cells (*Figure 2B and C*). However, the invariant cleavage pattern is affected in these embryos during the 4 to 8 cell mitosis which normally creates 2 layers of 4 animal and four vegetal (*Figure 2B and C*). For example, downregulating cell adhesion causes embryonic morphologies at the 8 cell stage ranging from wild type patterns (2 layers of 4 cells) to one layer of 8 cells with all intermediate patterns (*Figure 2B and C*). Computational analysis of the cell divisions occurring in the plane of imaging revealed a good prediction of spindle positioning by the computational mode (orienting deviation: 9,64 ± 4.16° in DA-aPKC expressing embryos, *Figure 2C*) suggesting that cell shape still regulates spindle orientation in these cells. Finally cell divisions at 16 and 32 cell stages were random giving rise to 64 cell blastula of highly variable morphologies (data not shown) preventing further manual analysis.

These observations show that cell shape can orient spindle positioning in early ascidian embryos and indicates that cell adhesion and regulated apicobasal polarity are absolutely vital to maintain the cell shapes supporting the invariant cleavage pattern of early ascidian embryos.

## The complete pattern of cell divisions of the ascidian blastula is predicted by a computational model based on apical cell shape at metaphase

Having revealed that mitotic spindles aligned with the long length of the apical surface during metaphase causing some mitotic spindles to rotate through approximately 90° while others remained relatively fixed in position, we wished to determine if apical cell shape can predict spindle orientation in the apical plane in all blastomeres up to the 64 cell stage. Defining the apical plane of an irregular polyhedral shape created by living cells is challenging since 2D imaging provides inherently false information as mitotic spindles in three dimensional embryos rarely align within the imaging plane. We therefore performed confocal 3D + time lapse imaging of living ascidian embryos stained with Cell Mask Orange (Invitrogen) to monitor cell membranes and spindle poles at metaphase in all blastomeres (see *McDougall et al., 2015* for a detailed protocol). The apical plane of each blastomere (defined as the 2D plane containing the poles of the mitotic spindle which can separate most of the apical surface from the basolateral surface) was systematically extracted from 3D-rendered blastomeres using Imaris software following a specific protocol (see Materials and methods and *McDougall et al., 2015*) for detailed protocols) and the position of the mitotic spindle in the extracted 2D plane was compared with the position of a mitotic spindle predicted by the computational model (*Minc et al., 2011*).

3D rendering and in silico isolation of each blastomere at interphase, prophase and metaphase revealed drastic cell shape changes during the cell cycle (*Figure 3*). Sphericity of each blastomere was determined using Imaris statistics (see Materials and methods) to estimate the complexity of polyhedral shape of in silico isolated blastomeres. Sphericity of blastomeres at metaphase was found to be significantly different from a round standard from the 4 cell stage and was particularly pronounced at the 16–24 and 32–44 cell stages during which spindle rotations occur (*Figure 3A*). At the 32 cell stage, in silico isolated blastomeres were found to be columnar during interphase and to flatten along the apical direction at metaphase (*Figure 3B*) without significantly changing cell volume (data not shown). This was reflected in an increase in the average sphericity of blastomeres at metaphase indicating that, at this stage, blastomeres partially round up at mitosis (*Figure 3B*). We then segmented manually apical (red) from basolateral (green) membranes on each 3D-rendered blastomere and calculated the apical surface ratio (apical surface divided by total surface) and found that a large increase in apical surface area accompanies mitotic cell rounding (from 0,15–0,23 at interphase to 0,42–0,56 at metaphase, *Figure 3B*) similarly to what is observed in differentiated epithelia (*Ragkousi and Gibson, 2014*). These observations document partial mitotic cell rounding at the sixth cell cycle which results in blastomeres increasing their apical surface by apical expansion at prophase and metaphase. Therefore the apical surface ratio is a better quantitative measure of cell shape changes than sphericity as some cell shape changes observed from a columnar cell (with a small apical surface) to a 'brick' shaped cell (with a larger apical surface) are not associated with changes in sphericity.

We then extracted the apical plane of each blastomere from the 2 cell stage to the 44 cell stage. The cell outline drawn from the 2D extracted apical plane of each blastomere was then computed to predict spindle orientation (*Figure 3C*, *Figure 4*). *Figure 4A* shows the observed and predicted spindle positions in the extracted apical plane of each blastomere from the fourth mitosis (8–16 cell) to the sixth mitosis (32-44-64 cell) and reveals a good prediction of spindle orientation by apical cell shape in almost all blastomeres. Quantification of the deviation between the predicted center of spindles and the observed center of spindles (centering deviation) from the 2 cell stage to the 44 cell stage confirms that all spindles are centered within 20% except for the B4.1 and B6.3 blastomeres of the germ cell lineage which undergo CAB-dependant unequal cleavages (*Figure 4B*). Strikingly we found that orienting deviation is under 30° in all cells except in B5.2/B6.3 and A6.3 (*Figure 4C*). By comparing the orienting deviation observed in 149 cell divisions (excluding the germ cell precursors that divide unequally) to randomly generated angles between 0 and 90° we found that the distribution of observed orienting deviations is non uniform and significantly different from a random distribution (*Figure 4D*). In our data set, 88% of cells show an orienting deviation of 30° or

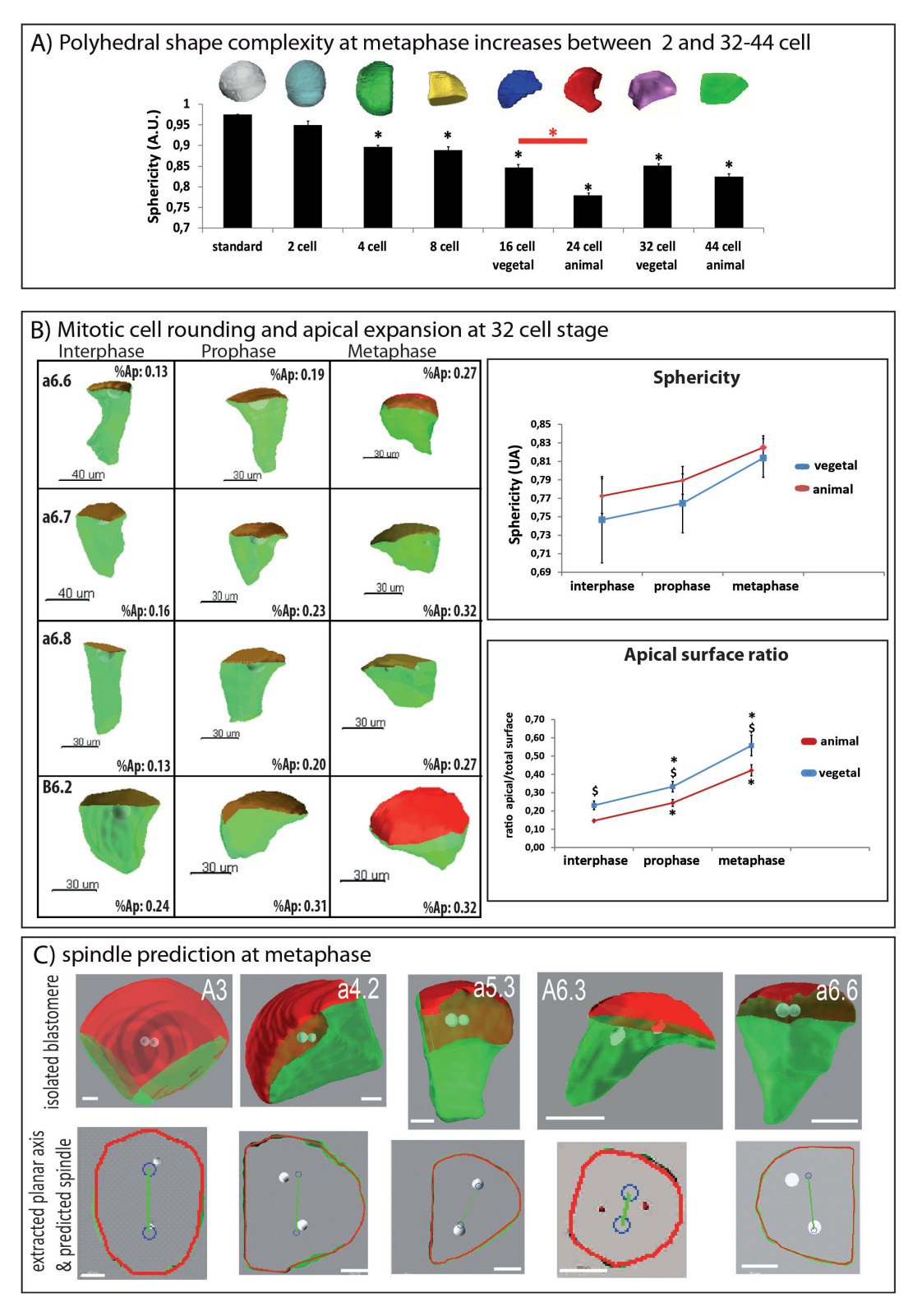

**Figure 3.** Changes in 3D cell shape during development and during the cell cycle. (**A**) Quantification of polyhedral shape complexity of in silico isolated blastomeres at metaphase (using sphericity measurements of Imaris software). While blastomeres at the 2 cell stage have a similar sphericity than a spherical standard (standard: 0.975+/0.001 (n = 4); 2 cell: 0.950 ± 0.010 (n = 4)), from the 4-cell-stage on sphericity significantly decreases compared to standard (p<0.05, black asterisk). Average values are: 4 cell: 0.897 ± 0.013 (n = 8); 8 cell: 0.889 ± 0.009 (n = 8); 16 cell (vegetal): 0.846 ± 0.006 (n = 8), 24

*Figure 3 continued on next page*

*Figure 3 continued*

cell (animal): 0.779 ± 0.007 (n = 8), 32 cell (vegetal): 0.851 ± 0.012 (n = 8); 44 cell (animal): 0.825 ± 0.012 (n = 8). Note that animal blastomeres (24 cell stage) have a significantly more complex polyhedral shape than their vegetal counterparts (16 cell) (p<0.05, red asterisk). An example of an in silico isolated blastomere is depicted above each bar of the graph. (B) Quantification of cell shape changes during the cell cycle at the 32–44 cell-stage. Left: 3D views of manually segmented blastomeres at interphase, prophase and metaphase (32 cell stage) showing cell shape changes between interphase, prophase and metaphase (inset: apical surface ratio of cell shown). Green is basolateral and red is apical. Scale bar as indicated. Top right: quantification of cell sphericity at interphase, prophase and metaphase. 6 blastomeres of the animal (red) and vegetal (blue) hemisphere of the 32 cell stage were averaged. The sphericity was significantly higher at metaphase than at interphase (black asterisk, p<0,05). Bottom right: quantification of the apical surface ratio at the same time points. The same blastomeres as in the sphericity graph were used to average apical surface ratio in the animal (red) and vegetal (blue) hemisphere. The apical surface ratio at prophase and at metaphase were significantly higher than at interphase (black asterisks, p<0,05). The apical surface ratio was higher in vegetal blastomeres than in animal ones (§ sign, p<0,05). (C) Pipeline for predicting spindle position using 2D computational model (*Minc et al., 2011*). See *McDougall et al. (2015)* for the full protocol of apical plane extraction. Top row shows examples of 3D rendered, in silico isolated blastomeres. Bottom row shows the extracted apical plane of the corresponding blastomeres with spindle predictions (blue circles joined by a green line). Scale bars = 20 μm.

less, 78% of cells have an orienting deviation under 20° and 52% of cells show an orienting deviation of less than 10° (*Figure 4D*). Therefore our model robustly predicts that spindle aligns with the long length of the apical surface in most cells with a precision of 30° and 20°. In contrast spindle orientation could not be predicted reliably in B5.2, B6.3 and A6.3 blastomeres (i. e. orienting deviation is above 30°) suggesting that in these three blastomeres apical cell shape does not regulate spindle positioning in the apical plane. This was expected in B5.2 and B6.3 which display spindle rotation during unequal cleavage (*Prodon et al., 2010*), but not in A6.3 which do not show spindle rotation nor unequal cleavage. However A6.3 cells undergo an asymmetric division segregating endoderm and mesoderm fates driven by asymmetric Ephrin and MAPK signalling (*Shi and Levine, 2008*) suggesting that, like in the germ lineage, cues other than apical cell shape might regulate spindle orientation in A6.3 blastomeres.

## Maternal and zygotic contributions to the stereotyped pattern of cell divisions

In the early ascidian embryo, unequal cleavages in the germ lineage are regulated by a maternal factor located in the posterior pole of the embryo called the Centrosome Attracting Body (CAB). The CAB is a cortical complex composed of cell polarity proteins (*Patalano et al., 2006*) which is segregated in the germ lineage and is responsible for spindle orientation in the B5.2/B6.3 mitoses in a cell autonomous manner (*Nishikata et al., 1999*; *Prodon et al., 2010*) against apical cell shape (*Figure 4C*). Conversely, zygotic transcription is required for germ layer patterning starting from the 16 cell stage in ascidians (*Lemaire, 2009*; *Hudson et al., 2013*) and also for generating cell cycle asynchrony from the 16 cell stage (*Dumollard et al., 2013*).

Removing the precursor of the CAB by dissecting the contraction pole (CP) at the zygote stage prevents unequal cleavage in the germ lineage as well as β-catenin stabilization in the endomesoderm giving rise to a synchronous hollow blastula that cannot gastrulate (*Nishida, 1996*; *Dumollard et al., 2013*). For example, removing the CP prevents unequal cleavage in CAB-containing blastomeres at the 8 cell stage (B4.1 pair) resulting in more centered spindles (*Figure 5B*) and the absence of small cells in the vegetal posterior pole of the embryo of CP-ablated embryos (*Figure 5A*). The general morphology of radialized embryos indicates that all cell shapes in the whole embryo are affected at the 16–32 cell stages (*Figure 5A*). Computational analysis of the cell divisions occurring in the imaging plane shows orienting deviations below 30° at 8, 16 and 32 cell stages (*Figure 5B*) suggesting that cell shape still regulates spindle orientation in CAB-depleted embryos. Therefore, the activity of the maternal CAB impacts not only unequal cleavage in the germ lineage but also cell shape and hence cell divisions in the rest of the embryo.

We then assessed the specific impact of zygotic transcription on the ascidian invariant cleavage pattern by either blocking all zygotic transcription using Pem1 expression (see *Kumano et al., 2011*; *Shirae-Kurabayashi et al., 2011* and *Dumollard et al. (2013)* for details on the effects of PEM1 ectopic expression) or by blocking β-catenin transactivation using DN-Tcf (which does not affect non transcriptional functions of β-catenin, see *Dumollard et al., 2013* for details). Embryo morphology

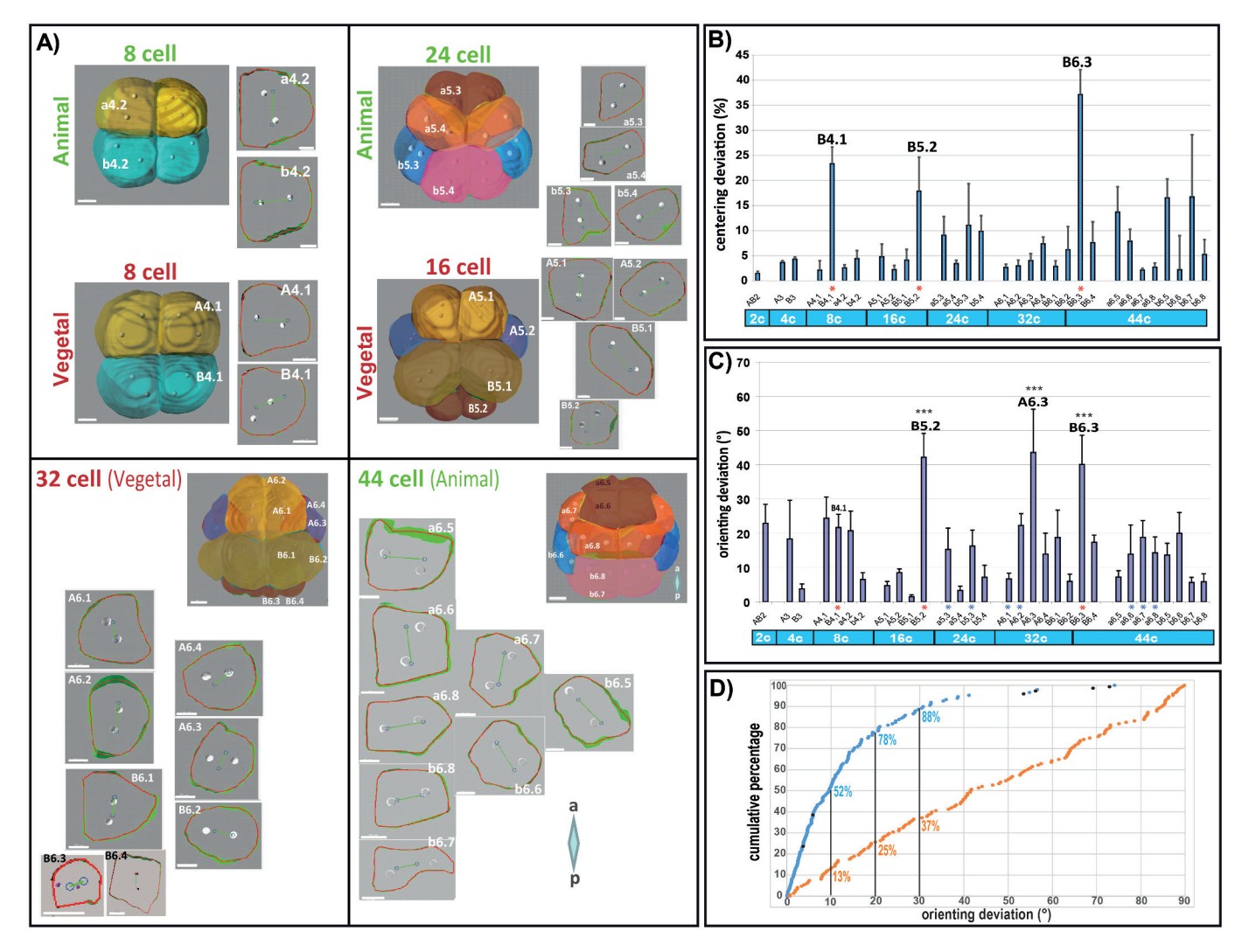

**Figure 4.** Computational model predicts spindle position and orientation in the apical plane of each blastomere. (**A**) 3D views of Phallusia embryos from 8 cell stage to 44 cell stage and extracted apical plane of each blastomere. Observed spindle poles are depicted by white circles/balls. Predicted spindles are depicted with blue circles joined by a green line. The red outline of each cell is the shape used by the computational model to predict spindle position. a=anterior, p=posterior, scale bars are all 20 μm. Lineages are color coded like in *Figure 1*. (**B**) Mean centering deviation in each blastomere. n = 6 cells analysed for each blastomere except A6.4 (n = 4); B6.1 (n = 5); a6.7 (n = 4), b6.5 (n = 4). Red asterisk denote cells undergoing unequal cleavage. (**C**) Mean orienting deviation in each blastomere. n = 6 cells analysed for each blastomere except A6.4 (n = 4); B6.1 (n = 5); a6.7 (n = 4), b6.5 (n = 4). Red asterisks denote cells undergoing unequal cleavage. Blue asterisks denote blastomeres undergoing OCD. Triple black asterisks denote that orienting deviation in the grouped B5.2, A6.3, and B6.3 cells are statistically greater than those in other lineages (Wilcoxon rank sum test with continuity correction, p-value=4.789*10$^{-7}$). (**D**) Quantification of orienting deviation: cumulative percentage graph of measured data (blue dots, n = 149 cell divisions, black dots denote the six A6.3 cells analysed) and randomly generated data (orange dots, n = 149). The measured data are not uniform and significantly different from the random data (One-sample Kolmogorov-Smirnov test, p<2.2*10$^{-6}$). The numbers indicated under each graph is the proportion of cells with orienting deviations under the considered threshold (10°, 20°, and 30°).

at 16 cell stage and CAB-dependant unequal cleavages are maintained in embryos expressing Pem1 ectopically as shown previously (*Negishi et al., 2007*; *Kumano et al., 2011*; *Dumollard et al., 2013*). However, all mitotic spindle rotations previously observed at the 16 and 32 cell stages are strongly impaired in Pem1 and DN-Tcf expressing embryos resulting in cell divisions with different orientation from the invariant cleavage pattern (termed 'misoriented cell divisions') (*Figure 6*). Most strikingly linear and T patterns of two sister cell spindles at the 32-44-64-cell mitosis were often

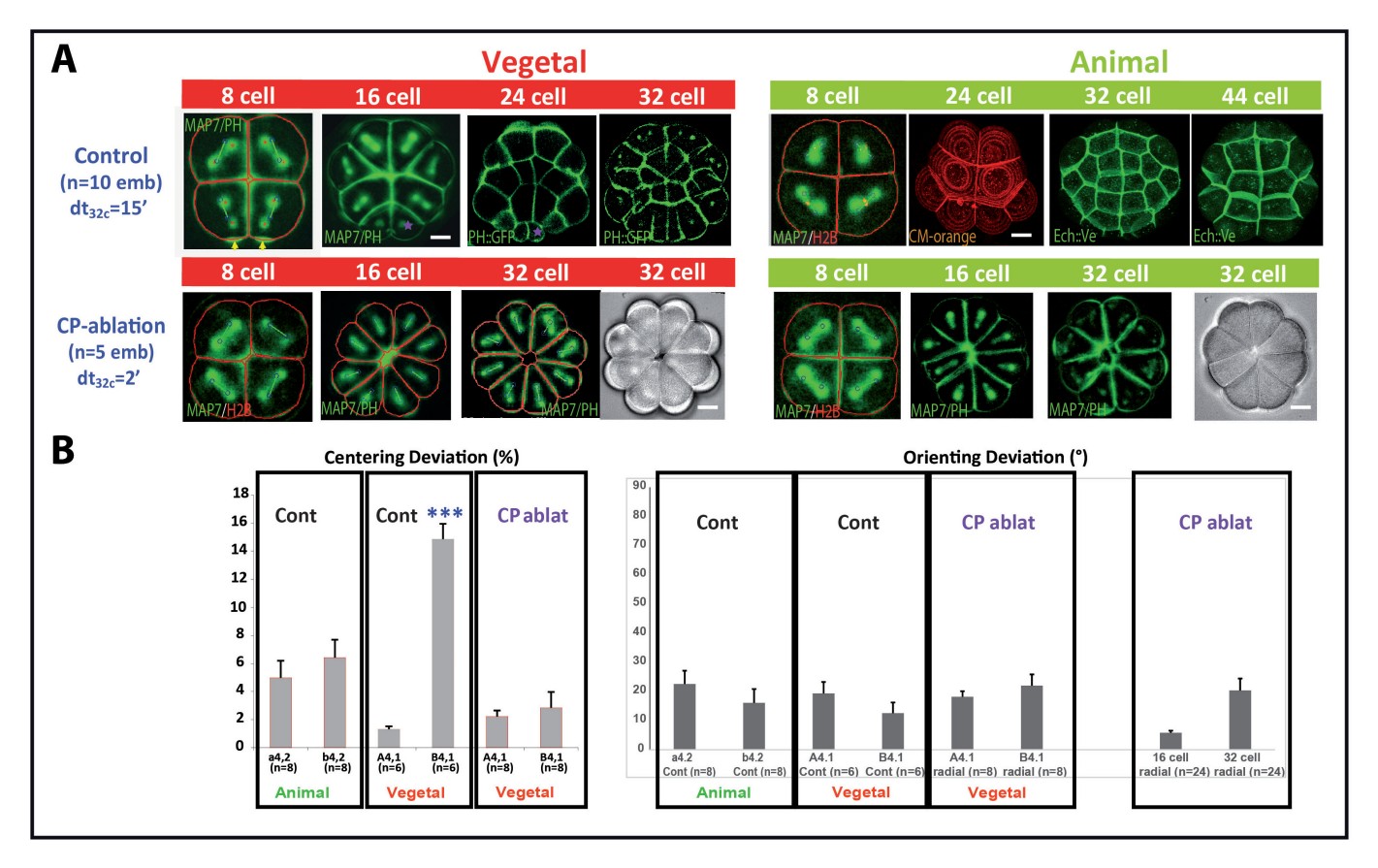

**Figure 5.** Removing the maternal CAB prevents unequal cleavage and radializes embryos. (**A**) Images showing embryonic morphology and cell shapes in Control (top row) and CP-ablated (bottom row) embryos. Top row: Images showing mitotic spindles (MAP7::GFP) and cell membranes (PH::GFP, CM-orange: Cell Mask Orange; Ech::Ve: Echinoid::Venus). Scale bar = 15 µm. Animal hemisphere (An) is depicted in green while the vegetal hemisphere (Veg) is depicted in red. Purple asterisks indicate unequal cell division in the germ line. Bottom row: images showing mitotic spindles and cell membranes (MAP7::GFP and PH::GFP) in radialised embryos in which the contraction pole (CP) was removed (CP-ablation). These embryos (n = 5) are completely radialised and do not bear small cells in the vegetal posterior pole of the embryo. Scale bar = 15 µm. (**B**) Left: Quantification of centering deviation showing that it is less than 10% in a4.2, b4.2 and A4.1 blastomeres whereas it is over 10% in B4.1 of control embryos. In CP-ablated embryos centering deviation is not affected in A4.1 but is decreased in B4.1. Triple asterisk indicates a significant difference with A4.1 (student, p=0.00004). Right: Quantification of orienting deviation showing that it is below 30° in both control and CP-ablated embryos at the 8 cell stage (a4.2, b4.2, A4.1, B4.1) and at the 16 and 32 cell stages in CP-ablated embryos.

replaced by two parallel spindles (giving rise to square pattern of cells) in these embryos (outlined in red in *Figure 6A*). Counting the occurrence of misoriented cell divisions in embryos expressing Pem1 or DN-Tcf (*Figure 6B*) shows that the blastomeres undergoing spindle rotation displayed more occurrence of misoriented cell divisions (a5.3, b5.3, A6.1, A6.2, a6.6, a6.7, a6.8 marked with an asterisk in *Figure 6B*). b6.5 and b6.6 blastomeres also showed a strong incidence of misoriented cell divisions but as a consequence of their mother (b5.3) being misoriented. The shape of the apical surface appears altered both in Pem1 and DN-Tcf expressing embryos and computational analysis of cell divisions in a6.8 (normally displaying spindle rotation) shows that misoriented cell divisions are still predicted reliably by apical cell shape (*Figure 6C*). Therefore the processes supporting spindle orientation in the long length of the cell in the apical plane are maintained in these embryos and misoriented cell divisions are probably due to altered cell shape in manipulated embryos.

Since both PEM1 overexpression and DN-Tcf reduce cell cycle asynchrony (*Dumollard et al., 2013*) we next sought to directly alter cell cycle duration to assess the impact cell cycle asynchrony has on the invariant cleavage pattern.

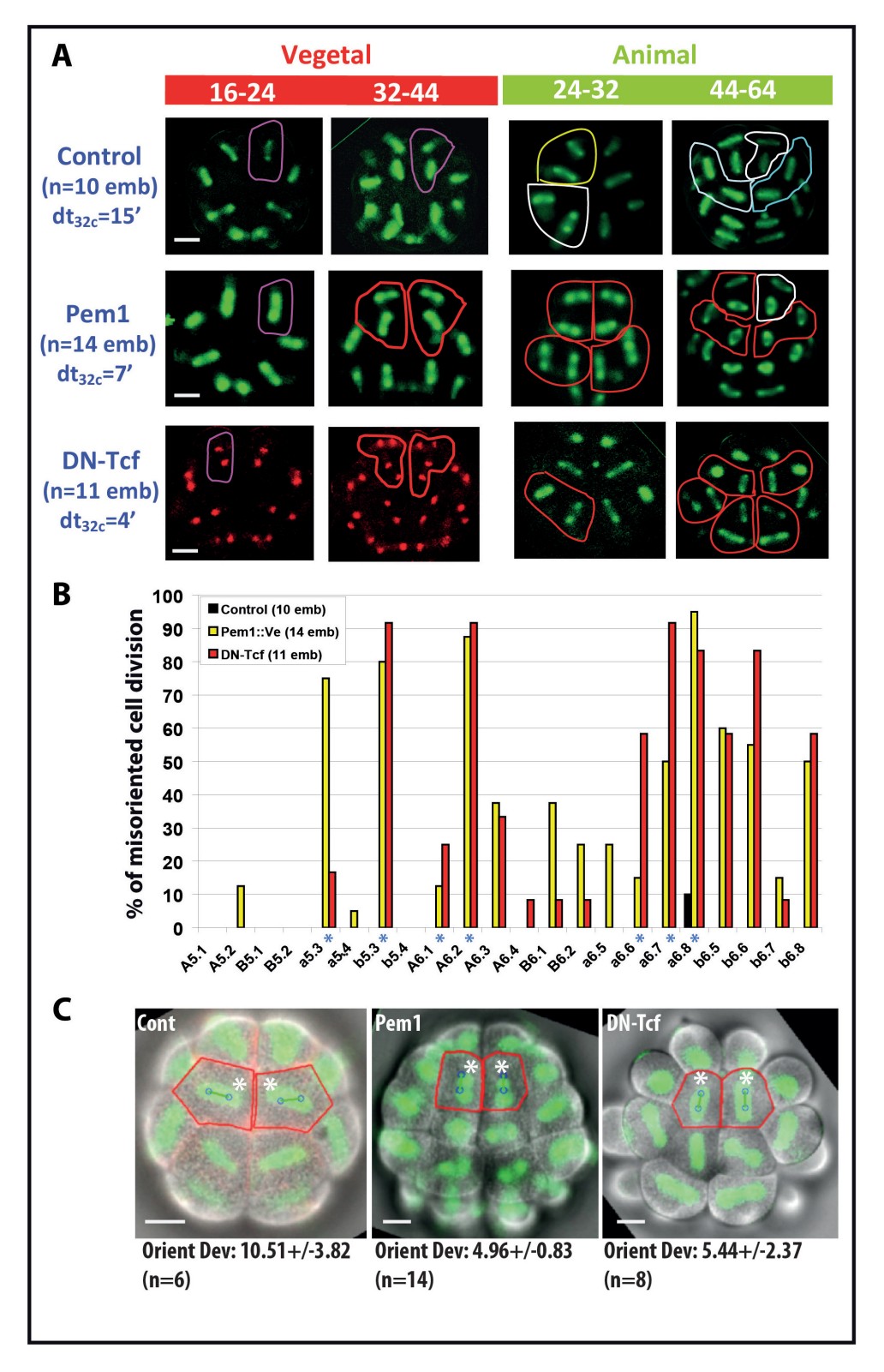

**Figure 6.** Impact of zygotic transcription on the invariant cleavage pattern. (**A**) Images showing metaphase spindle (green) and nuclei (red) in control, Pem1::Ve and DN-Tcf expressing embryos. Unaffected cell divisions are surrounded by a colored line (in control and manipulated embryos). Misoriented cell divisions are surrounded by a red line. Scale bars = 20 μm. dt32c indicates the difference in timing of mitotic entry between animal and vegetal hemispheres at the 32–44 cell stage. (**B**) Graph plotting the incidence of misoriented cell divisions in control (black bars), Pem1::Ve (yellow bars)

*Figure 6 continued on next page*

*Figure 6 continued*

and DN-Tcf (red bars) embryos. Blue asterisks denote blastomeres undergoing OCD. (**C**) Images showing embryonic morphology in the ectoderm (animal) at the 32–44 cell stage. Blastomeres with asterisks are a6.8 for which cell outline (in red) and spindle prediction (blue circles joined by a green bar) are depicted. Orienting deviation in a6.8 blastomeres displaying OCD in control embryos and misoriented cell divisions in a6.8 blastomeres in Pem1 and DN-Tcf embryos are indicated under the images. n indicates the number of a6.8 blastomeres analysed. Scale bars are 20 μm.

## Inhibiting cell cycle asynchrony disrupts the invariant pattern of cell divisions

In Xenopus, Drosophila or Zebrafish embryos early embryonic cell cycles are regulated by a balance of the two cell cycle regulators Wee1 and Cdc25 which inhibit or activate cdk1 respectively to set interphase length (reviewed in *Farrell and O'Farrell, 2014*). Wee1 maternal RNAs are present in the

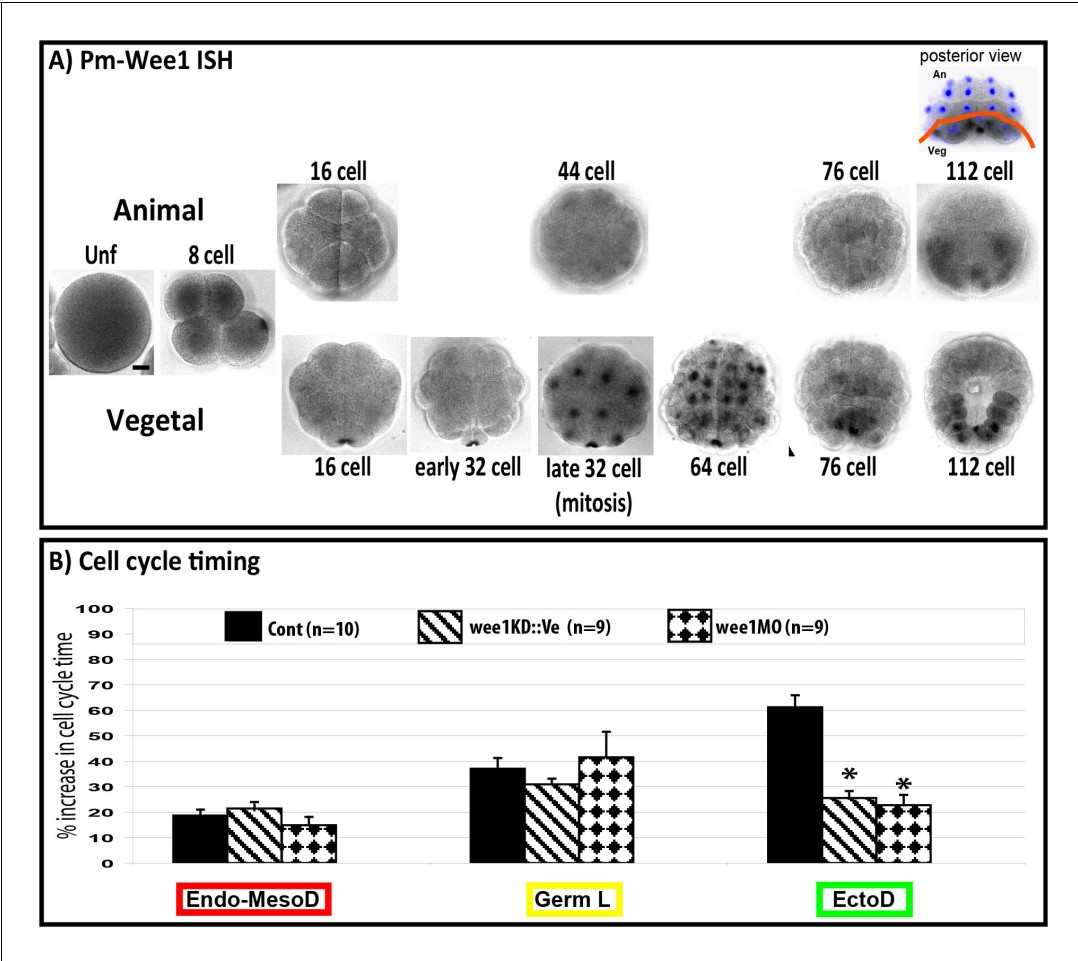

**Figure 7.** Inhibiting cell cycle asynchrony in the ascidian blastula. (**A**) Images showing in situ hybridizations of Pm-Wee1 from the unfertilized egg (Unf) to the late 112 stage. Wee1 is expressed in the unfertilized oocytes where it is slightly enriched in the vegetal cortex of the egg. Then a maternal signal can be observed in the early stages with a specific enrichment in the CAB region hosting the germ plasm. Zygotic Wee1 is expressed in the endoderm precursors from the late 32 cell stage to the 76 cell stage and in the muscle precursor from the 112 cell stage (i.e., just before they gastrulate). In some images Hoechst staining of nuclei is shown in blue. Scale bar is 20 μm. (**B**) Quantification of cell cycle length at the 16–32 cell stage (MBT, cell cycle 5) in manipulated embryos: control embryos (black bars, n = 10), embryos expressing Wee1KD::Ve (stripped bars, n = 9), embryos injected with wee1 MO (diamond bars, n = 9). wee1KD and wee1 MO both speed up the ectoderm cells (EctoD) without affecting the endomesoderm (Endo-MesoD) or the germ line (Germ L). % increase in cell cycle time relative to cell cycle timing at the 8 cell stage. A 20% increase means that the cell cycle timing has increased by 20% compared to the previous cell cycle at 8 cell stage (see *Dumollard et al., 2013* for details). Triple black asterisks indicate p=0.0003 for wee1KD and p=0.000009 for wee1MO.

unfertilized oocyte in Phallusia mammillata (*Figure 7A*, see also RNA-seq data in Aniseed database: http://www.aniseed.cnrs.fr/ with Gene Id: phmamm.CG.MTP2014.S423.g08568) where it is enriched in the vegetal cortex. Then maternal Wee1 transcripts can be observed in the CAB similarly to PEM-type genes (*Prodon et al., 2007*). Zygotic Wee1 transcripts first appear at late 32 cell stage in the endoderm lineage (64–76-stage) just before these cells invaginate. After the onset of gastrulation (112 cell stage) Wee1 transcripts are expressed in mesodermal cells just before they invaginate (*Figure 7A*). We modified Wee1 activity in the whole embryo (see Materials and methods, *Murakami et al., 2004*; *Farrell and O'Farrell, 2014*) and found, as anticipated, that Wee1 inhibition speeded up the slow cell cycle in the ectoderm at the 16 cell stage without affecting the cell cycle duration of the vegetal endomesoderm thereby eliminating the transient 24 cell stage (*Figure 7B*). At the 32 cell stage, cell cycle asynchrony between ectoderm (animal) and endomesoderm (vegetal) was 6–8 min in Wee1-manipulated embryos compared to 15 min in control embryos (*Figure 8A*). Analysis of cell division orientation at the 16–24 and 32–44 cell stages in Wee1-manipulated embryos revealed deviations from the invariant cleavage pattern as well as variability between individuals (*Figure 8A and B*). In synchronized embryos, misoriented cell divisions are particularly obvious at the 32–64 cell stage where the linear arrangements of A6.1/A6.2 and a6.8/a6.7 daughter cells observed in control embryos were changed to square arrangements of cells (circled in red in *Figure 8A*). Counting the occurrence of misoriented cell divisions revealed that the blastomeres b5.3, A6.1, A6.2, a6.7 and a6.8 were the most affected (*Figure 8B*). The orientation of cell division in b6.5 and b6.6 was also affected (*Figure 8B*). Therefore the blastomeres most affected by synchronization were the ones displaying oriented cell division (b5.3, A6.1, A6.2, a6.7 and a6.8) or daughters of cells displaying OCD (ie b6.5, b6.6). Apical surface area at metaphase was altered in Wee1-manipulated embryos and misoriented cell divisions in a6.8 blastomeres still aligned with the long length of the apical surface (orienting deviation: 10.17 ± 4.15°, *Figure 8B*). Therefore, inhibition of cell cycle asynchrony did not seem to disrupt the mechanisms supporting spindle orientation in the apical plane and misoriented cell divisions may rather be due to altered apical surface area.

## Discussion

During ascidian early embryogenesis, all planar cell divisions generating the blastula-stage embryo occur in a highly predictable manner which is implicit in the term 'invariant cleavage pattern'. In the absence of either cell migration or death, cell division orientation thus defines the precise topographical positioning of all cells in the whole blastula-stage embryo. Moreover, these precise cell positions are invariant between distantly-related species of ascidian. We find here that a small number of maternal factors and gene-regulatory networks (GRN) influence the final position of cells in early ascidian embryos. One maternal mechanism is provided by the CAB which influences the shape of all cells in the embryo indirectly since its ablation causes embryos to become radialized (hence affecting the shape of all cells in the embryo). One zygotic GRN that influences cell position is controlled by $\beta$-catenin which induces cell cycle asynchrony from the 16 cell stage to fine-tune cell position at the 24-32-44 cell stages. Finally, one additional maternal cue is provided by the polarisation of the cells which occurs during cleavage divisions and causes mitotic spindles to align parrallel to the apical cell surface. All of these cues/inputs are likely propagated between the cells of the embryo through cell adhesion and the resulting adhesive forces balanced with physical forces such as surface tension giving rise to the particular cell shapes observed. In particular, we report here that it is the long axis of the cell in the apical plane that is important for creating the invariant cleavage pattern.

### Apical cell shape supports OCDs and underpins the invariant cleavage pattern

The ascidian embryo provided a convenient experimental system in which to test the contribution of apical cell shape to spindle orientation by a computational mathematical model (*Minc et al., 2011*; *Minc and Piel, 2012*). We demonstrate here for the first time that a partial mitotic cell rounding accompanies cell division in the ascidian blastula whereas it was not observed in Xenopus (*Strauss et al., 2006*) and is still not clearly documented in Zebrafish (*Xiong et al., 2014*). Interestingly, in these two species, cell deformations have been observed at MBT but only in isolated blastomeres (called 'cell motility' in *Newport and Kirschner, 1982* and *Kane and Kimmel, 1993*).

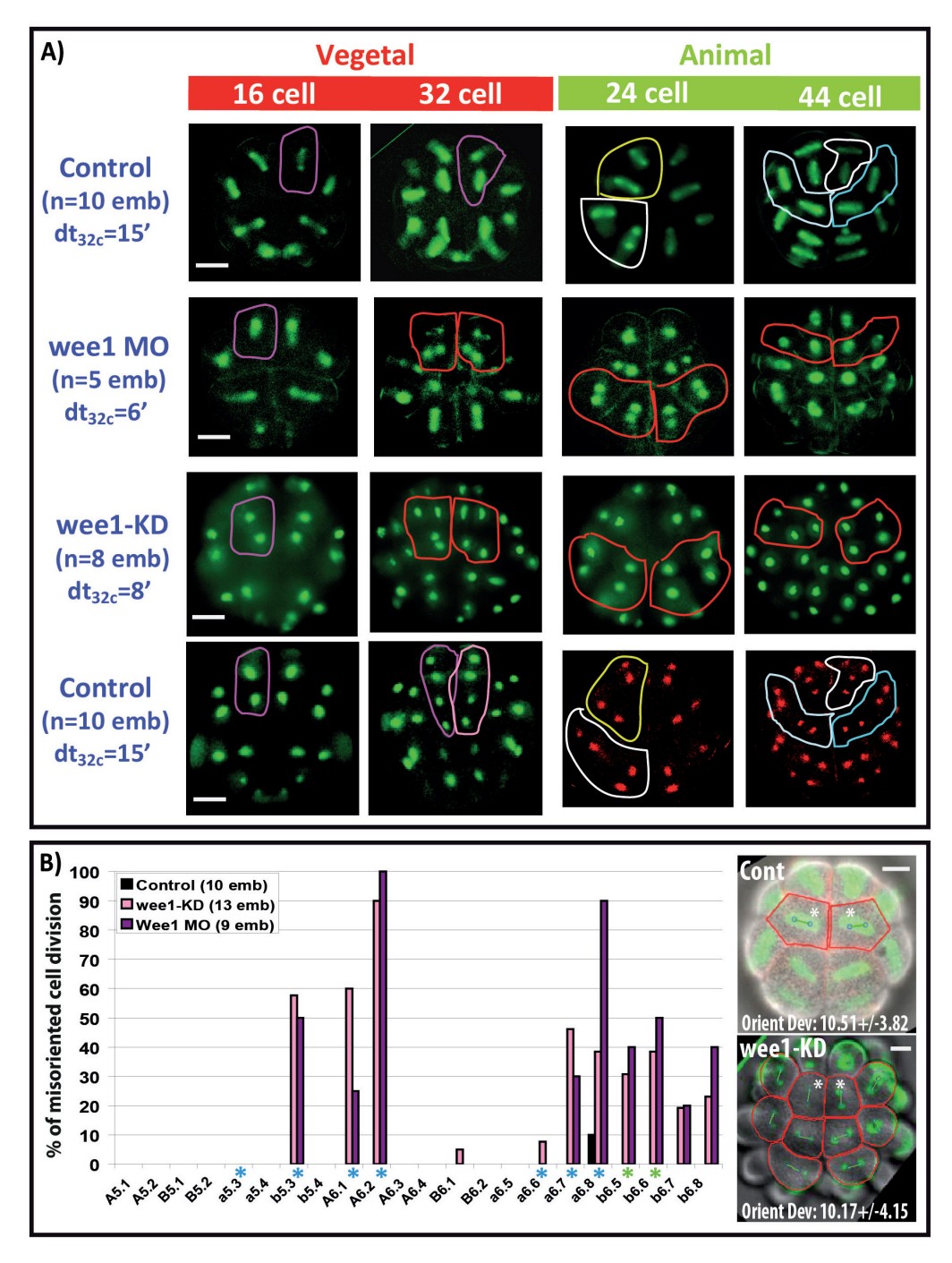

**Figure 8.** Impact of cell cycle asynchrony on the invariant cleavage pattern. (**A**) Images showing metaphase spindles in control and wee1 MO injected embryos (top two rows) or showing nuclei in wee1KD::Ve and control embryos (bottom two rows). Unaffected cell divisions are surrounded by a colored line in control embryos. Misoriented cell divisions occurring in manipulated embryos are surrounded by a red line. Scale bar = 20 μm. dt32c indicates the difference in timing of mitotic entry between animal and vegetal hemispheres at the 32–44 cell stage. (**B**) Graph plotting the incidence of misoriented cell divisions in control (black bars), wee1KD (pink bars) and wee1 MO (purple bars) injected embryos. Blue asterisks denote blastomeres undergoing spindle rotation, green asterisks indicate daughters of b5.3 (undergoing spindle rotation). Inset: Images showing embryonic morphology in the ectoderm (animal) at the 32–44 cell stage. Blastomeres with asterisks are a6.8 for which cell outline (in red) and spindle prediction (blue circles joined by a green bar) are depicted. Orienting deviation in a6.8 blastomeres displaying OCD in control embryos (n = 6 cells) and misoriented cell divisions in wee1-KD (n = 8 cells) embryos is indicated in the images. Scale bars are 20 μm.

However, even though the isolated blastomeres with the longest cell cycle showed more pronounced cell deformations, cell cycle dependence of such cell deformations remains unclear. Mirror-like cell shape changes in 32–44 cell stage ascidian embryos were noted previously (*Tassy et al., 2006*), prompting us to test the idea that cell cycle asynchrony may be involved in the mirror-like cell shape regulation. We demonstrate here that these cell shape changes consist in cell rounding and apical expansion during mitosis and they correspond to the mitotic cell rounding observed in epithelial cells (*Ragkousi and Gibson, 2014*). In contrast to cells in differentiated Drosophila epithelia which completely round up at mitosis (*Bosveld et al., 2016*) but similarly to the epibolizing enveloping cell layer (EVL) of Zebrafish embryos (*Campinho et al., 2013*), mitotic cell rounding in the ascidian blastula is incomplete. In consequence, blastomeres at the 16–24 and 32–44 cell stage do not become completely round and the shape of the apical surface remains anisotropic during metaphase.

After confirming in ascidian embryos that, like in C elegans, sea urchin, Zebrafish or Xenopus embryos (*Wildwater et al., 2011*; *Minc et al., 2011*; *Strauss et al., 2006*; *Pierre et al., 2016*), cell shape impacts the orientation of cell division, we show that a computational model based on the shape of the apical surface predicts spindle orientation in the apical surface in all cells of the embryo except in the germ lineage (B5.2, B6.3) and A6.3. 88% of these cells can be predicted by the model with a 30° precision and 78% of these cells are still predicted by the computational model with 20° precision (*Figure 4D*). These observations strongly indicate that mitotic spindles align with the long axis of the cell in the apical plane to implement the invariant cleavage pattern of planar cell division thus creating the distinctive topographical organization of cells in the ascidian blastula. In the ascidian blastula such a general geometric rule is overridden only in the germ lineage where a maternally derived cortical polarity complex (the CAB) attracts one spindle pole duing unequal cell division (*Prodon et al., 2010*). It would be interesting to extend this mathematical analysis to other developmental stages and to distantly-related ascidian species, since we may have unveiled a conserved apical cell shape dependent mechanism underpining the invariant cleavage pattern displayed by all ascidians.

By following the position of clones of cells from the 16 cell stage to the 64 cell stage we inferred the occurence of OCDs (when daughter cell spindle is not orthogonal to the spindle orientation of its mother) in specific cells. By imaging spindles during mitosis in these cells we discovered that they all perform OCD through spindle rotation rather than migration/rotation of the nucleo-centrosomal complex. A6.1 is the only exception to this observation where it is the nucleo-centrosomal complex rather than the spindle that rotates, but OCD in this lineage still depends on apical cell shape (see *Negishi and Yasuo, 2015* for details). Thus we argue that, together with apico-basal polarity which likely enforces the planar orientation of cell division, apical cell shape is a major driver of cell positioning in the ascidian blastula. This hypothesis implies that apical cell shape at metaphase must be very stereotyped and conserved in ascidian early embryos. Although perhaps surprising, this finding should be considered with the fact that cell position in ascidian embryos at the 64 cell stage is crucial since it is at this time that the fate map is established (*Lemaire, 2009*). More specifically, precise cell position in the anterior part of the embryo determines the number of animal marginal cells (a6.5, b6.5) that will receive a local cell induction from contacting vegetal cells to create a neural ectoderm of 6 cells at the 64 cell stage (*Lemaire, 2009*). Mesodermal cells of the vegetal hemisphere located in the marginal region segregate, during mitosis 32–44 cell, neural and notochord fates in the anterior part of the embryo (A6.1, A6.2) and muscle and mesenchyme (B6.2) fates in the posterior part of the embryo (*Kumano and Nishida, 2007*). Therefore, the invariant cleavage pattern and oriented cell divisions might be part of the mechanism enforcing fate segregation in the ascidian blastula.

## Cell cycle asynchrony underpins the invariant spatial pattern of cell divisions

A central finding of this study is the causal relationship between cell cycle asynchrony and the orientation of cell division. We suggest that cell cycle asynchrony impacts the spatial pattern of planar cell divisions by regulating the shape of the cell's apical surface at metaphase. First, we found that every mitotic spindle tends to align with the long length of the cell's apical plane at metaphase. Second, abolishing the asynchrony that causes the appearance of the 24 cell stage altered the invariant cleavage pattern. Third, misoriented cell divisions in synchronized embryos are still reliably predicted by apical cell shape.

Critically, either causing vegetal cell cycles to become slower at the 16 cell stage by inhibiting zygotic transcription, or making animal cell cycles faster by inhibiting Wee1 kinase activity both had the same overall effect: all cells divided synchronously and the invariant cleavage pattern was disrupted. We conclude that overall cell cycle duration is not important, and rather it is the asynchrony between the animal and vegetal halves of the embryo that is crucial. Since ascidian embryos live in a marine environment that does not have a constant temperature, absolute cell cycle duration is likely less important than the cell cycle asynchrony which is maintained over a range of tempartures in different species of ascidian. Therefore the zygotic GRN driven by nuclear $\beta$-catenin that patterns germ layers is also responsible for causing cell cycle asynchrony, which in turn enforces the invariant cleavage pattern through cell shape dependent mitotic spindle orientation in the apical plane at the 16-32-44-cell stages.

We find here that entry into mitosis is accompanied with partial cell rounding via apical expansion (and hence a reduction in cell adhesion) in the ascidian blastula. It was found in Ciona embryos that 'between the early 32-cell stage and the mid 44-cell stage, the elongation factors of opposing animal and vegetal cells change in precisely opposite manner with time' (*Tassy et al., 2006*). We hypothesize that such mirror behaviors between animal and vegetal blastomeres is brought about by asynchronous mitotic cell rounding between animal and vegetal blastomeres. Precise shape of the apical surface during mitosis may be a function of not only adhesion with neighbouring cells but also of the remaining adhesion between interphasic and mitotic cells in opposite hemispheres since during cell division in ascidians the adhesion between blastomeres remains. We found that apical cell shape was altered in animal cells when we slowed down the cell cycle of vegetal cells with DN-TCF or PEM1 (*Figure 6*). Likewise, we report that apical cell shape was altered in vegetal blastomeres when we speeded up the cell cycle of animal cells by inhibiting Wee1 (*Figure 8*). We therefore conclude that the shape of the apical surface of animal cells is affected by the cell cycle state of the vegetal cells, and likewise that the shape of the apical surface of vegetal cells is affected by the cell cycle state of the animal cells. It is interesting to note that the overall cell cycle asynchrony between the animal and vegetal cells is about 15 min. which is about the duration of M phase. We wonder whether this may be one of the selective pressures leading to the retention of the cell cycle asynchrony between distantly-related ascidians. Indeed, both phlebobranchs and stolidobranchs ascidians display nuclear $\beta$-catenin in vegetal cells at the 16 cell stage (Ciona: *Hudson et al. (2013)*, Halocynthia: *Kawai et al., 2007*), and it is a GRN controlled by $\beta$-catenin that causes cell cycle asynchrony starting at the 16 cell stage (*Dumollard et al., 2013*). It will therefore be interesting to elucidate the entire GRN that controls cell cycle duration in the ascidian at the 16–24 cell stage, and to determine how conserved that GRN is between distantly-related ascidian species.

Importantly the removal of the axial determinant (the pre-CAB or centrosome-attracting body) that generates unequal cell division in the germ lineage (*Nishikata et al., 1999*; *Patalano et al., 2006*) not only prevents unequal cleavage in the germ lineage but also affects cell division orientation in the whole embryo as CAB-ablated ascidian embryos are completely radialized (*Nishida, 1996*, this study). Unequal cleavage of the two vegetal posterior blastomeres at the 16 cell stage thus affects the shape of every cell in the early embryo. Such effect of unequal cleavage on cell division of distant cells is supported by cell adhesion-dependant mechanical coupling between blastomeres which was found to be necessary to maintain the invariant cleavage pattern. It is noteworthy that regulated apicobasal polarity is crucial to maintain cell adhesion in the ascidian embryo to propagate individual cell deformations to the rest of the embryo and implement the invariant cleavage pattern.

Given the important role played by the shape of the apical surface in spindle orientation in ascidian early embryos it is evident that physical cellular properties that minimize energy during cell packing likely play an important role in cell division plane specification. Further studies will be required to understand how the apical surface of every blastomere is interdependent on neighboring cells due to packing constraints. Such interdependence between cell division plane orientation and apical cell shape is involved in a number of morphological processes. In vertebrate embryos oriented tissue strain generated by the gastrulating mesoderm determines the global axis of planar polarity in the Xenopus ectoderm (*Chien et al., 2015*), while cell division is oriented by tissue tension in the Zebrafish ectoderm to improve epithelial spreading over the yolk layer during epiboly (*Campinho et al., 2013*; *Xiong et al., 2014*). In ascidian blastulae, we hypothesize that the shape of cells in mitosis is a function of the tension generated between the dividing cells and their neighboring interphasic cells coupled with the tension between the cells that are dividing. This may exert stereotyped

deformations of adhering mitotic cells to generate the invariant cleavage pattern. However, further studies are needed to assess whether global tissue tension deforms blastomeres of the ascidian embryo or whether, on the contrary, autonomous cell cycle-driven cell shape changes are transmitted in the embryo via cell adhesion.

## Materials and methods

### Biological material

Eggs from the ascidians Phallusia mammillata were harvested from animals obtained in Sète and kept in the laboratory in a tank of natural sea water at 16°C. Egg preparation and microinjection have been described previously (see detailed protocols in *McDougall et al., 2014*, *2015*). All imaging experiments were performed at 19°C.

### Live imaging of Phallusia embryos

Microtubules and mitotic spindles were imaged using our characterised constructs of either MAP7::GFP or Ensconsin::3XGFP (*McDougall et al., 2015*). DNA and nuclei were imaged with H2B::mRfp or the nuclear proteins (wee1KD::Ve, Ve::cdc45, *Dumollard et al., 2013*). Plasma membrane was stained using PH::GFP or PH::dTomato or Cell Mask Orange (Invitrogen, see protocols in *McDougall et al., 2015*). Control embryos in *Figure 2A* are embryos stained with cell mask and cultured in filtered sea water (FSW). Control embryos in *Figures 2B,C*, *5*, *6*, *7* and *8* are embryos injected with cRNAs coding for MAP7::GFP or Ens::3XGFP or PH::GFP or Ve::cdc45 (in green) or H2B::mRfp (in red) or a combination of two of these markers. We have found that all these staining procedures have no impact on the invariant cleavage pattern (*McDougall et al., 2015*).

### Manipulation of zygotic transcription and of cell cycle timing in Phallusia embryos

To inhibit embryonic patterning Pm-Pem1 (*Shirae-Kurabayashi et al., 2011*; *Kumano et al., 2011*) and DN-Tcf (kindly provided by Yasuo Hitoyoshi (UMR7009, LBDV)) were used exactly as in *Dumollard et al. (2013)*. Ci-wee1 (gene Id: KH.S256.1) was amplified from a Ciona intestinalis Gateway-compatible cDNA library using PCR (*Roure et al., 2007*). To speed up cell cycle the activity of the Wee1 kinase was inhibited using a kinase dead form of Wee1 (Wee1KD) which was generated by introducing a stop codon inside the catalytic domain of the protein (resulting in a deletion of aa 532 to 633). Such a construct was shown to have a dominant negative effect on endogenous wee1 (*Murakami et al., 2004*). Alternatively, a morpholino target to Pm-wee1 (CAGGACCATATAAACTCCTACTGCT) was injected to decrease wee1 activity in the whole embryo. All constructs were made using pSPE3 (*Roure et al., 2007*) and the Gateway cloning system (Invitrogen) unless otherwise stated (see *McDougall et al., 2014*, *2015* for detailed protocols). Synthetic RNAs were injected in unfertilized eggs or in one blastomere of a 2 cell stage embryo.

### Manipulation of cell adhesion and apicobasal polarity and embryo compressions

To remove cell adhesion, Phallusia zygotes stained with Cell Mask-Orange or expressing MAP7::GFP and PH::GFP (to image cell membranes and spindle poles) were cultured in $Ca^{2+}$-free sea water (supplemented with 1 mM EDTA, as described in *Sardet et al., 2011*) and imaged up to the 64 cell stage.

Dominant active aPKC: Pm-aPKC (GenBank: AY987397.1, *Patalano et al., 2006*) was cut at K146 to remove the N-terminal regulatory domain of aPKC and tagged with Venus using our Gateway cloning system (*McDougall et al., 2015*; *Roure et al., 2007*). Removal of the N-terminal regulatory domain of aPKC results in a constitutively active form (DA-aPKC). This construct can expand the apical domain of superficial cells at the expense of the basolateral domain in Xenopus embryos (*Sabherwal et al., 2009*) and could significantly reduce cell adhesion in ascidian embryos. Ve::Tpx2 was expressed together with DA-aPKC in order to monitor spindles during mitosis (*McDougall et al., 2015*).

4 cell stage embryos stained with Cell Mask-Orange were compressed between slide and cover-slip. Only embryos showing a compressed Animal-Vegetal axis of 45 µm (confirmed by confocal imaging) or less were used for analysis.

## Time-lapse and fluorescence microscopy

Time-lapse imaging of Venus, GFP, mRfp1,Cherry and Tomato constructs was performed on a Zeiss Axiovert200 and a Zeiss Axiovert100 inverted microscopes set up for epifluorescence imaging. Sequential brightfield and fluorescence images were captured using a cooled CCD camera (Micromax, Sony Interline chip, Princeton Instruments, Trenton NJ) and data was collected using MetaMorph software (Molecular Devices, Sunnyvale CA) essentially as described in *McDougall et al. (2014)*, *(2015)*. Time series were reconstructed and analysed by MetaMorph and Image J (NIH, USA) software packages. 4D confocal imaging was performed on Leica CSLM SP2 through a long distance 40X (NA = 0.8) objective to obtain 3D embryos over time (30–35 z-planes imaged every minute) that were manually segmented and 3D rendered using Imaris 3.7. 2D imaging was performed for *Figures 1C*, *2*, *5*, *6* and *8* and only the cells whose spindle remains in the imaging plane during the whole of mitosis are analysed. 3D confocal imaging and 3D rendering were performed only for *Figures 3* and *4*.

## 3D rendering and plane extraction using imaris software

The complete protocol for 3D rendering of confocal stacks using Imaris (x64, version 7.7.2, Bitplane) is published in details in *McDougall et al. (2015)* and can be downloaded from http://www.biodev.obs-vlfr.fr/~dumollard/protocols/Segmentation-manuelle-Imaris-En.pdf. All cell contours were drawn manually on 2D-slices before 3D rendering and analysis by Imaris. Sphericity of the 3D shapes was calculated by Imaris software ('statistics' function) for each blastomere and compared to the sphericity of a spherical standard (an in vivo isolated blastomere at metaphase from a 2 cell stage which showed a sphericity of $0.975 \pm 0.001$, n = 4 cells). Sphericity was found to be significantly lower than spherical standard at the 4 cell stage ($0.897 \pm 0.013$, n = 8 cells). To calculate the apical surface ratio (apical surface related to total surface), the contact-free surface (apical in red) and the surface of contacts with other cells (basolateral in green) were manually segmented by cutting and duplicating the 3D surfaces of each blastomere (the protocol used for cutting 3D objects using Imaris can be downloaded from http://www.biodev.obs-vlfr.fr/~dumollard/protocols/protocole-surface-apicale-En.pdf). In *Figure 3B*, the apical surface ratio and sphericity of the eight quasi-synchronous animal cells was compared to the six quasi-synchronous vegetal cells at early interphase, prophase and metaphase. B6.3 and B6.4 were not analysed in this experiments because they were delayed with the rest of the embryo (by ~10 min).

The protocol for apical plane extraction in each blastomere is published in *McDougall et al. (2015)*. Briefly, the apical plane (i.e. the plane parallel to the apical surface) considered as the 2D plane comprising the two spindle poles which can separate most of the apical and basolateral membranes was cut (using clipping plane function of Imaris) from 3D rendered blastomeres.

## Spindle position prediction using computational model

Prediction of spindle position in 2D extracted planes was performed with MatLab using scripts which may be dowloaded at: http://www.minclab.fr/research/ (*Minc et al., 2011*). This model postulates that MTs radiating from spindle poles reach the cell cortex and pull with forces that scale to MT length. The model assays all possible orientations in the 2D shape and computes the evolution of the torque with spindle orientation, which informs on the mechanical equilibrium corresponding to stable spindle orientation. To implement the model with the current study, the outline of the cells and the experimental positions of the spindle poles were drawn manually as inputs. The script returned the difference between the centers of observed and predicted spindles (computed as a 'centering deviation' in %) and the difference in angle orientation between predicted and observed spindles (computed as an 'orienting deviation' in degrees (°)). A protocol for how this script was used in this study may be downloaded from http://www.biodev.obs-vlfr.fr/~dumollard/protocols/Minc-prediction-En.pdf .

## In situ hybridization on whole embryos

For in situ hybridization of mRNAs, embryos were fixed in 100 mM MOPS pH7.6/0.5 M NaCl/4% formaldehyde ON at 4°C and then washed in PBS, dehydrated in ethanol and stored at −20°C. Fixed embryos were then processed as described in *Sardet et al. (2011)*.

## Statistical methods

Bar graphs in all figures except in *Figures 6B* and *8B* show mean with error bars indicating s.e.m. The number of cell analysed for each graph bar is indicated in the figure legend. Statistical difference was evaluated by an unpaired two-tailed Student t-test (with Excell) and the Wilcoxon rank sum test, or the one sample Kolmogorov-Smirnov test with R. software package, and P values are depicted in the figure legends. Bar graphs in *Figures 6B* and *8B* show percentage of misoriented cell divisions (i.e. cell divisions showing a different orientation to the invariant cleavage pattern) in each blastomere calculated as the ratio of misoriented cell division divided by the total number of cell divisions analysed (indicated in the graph legend).

## Acknowledgements

We would like to thank the ANR (ANR-12-BSV2-0005-02) and ARC (n° SFI20111203776) for financial support. We would also like to thank Hitoshi Yasuo for helpful discussions and for providing WT $\beta$-catenin and DN-Tcf constructs. We thank JO Irisson (LOV, UMR 7093, CNRS) for his help with the statistical analysis shown in *Figure 4*. Finally, we thank the members of UMR7009 for their help and support.

## Additional information

### Funding

| Funder | Grant reference number | Author |
|---|---|---|
| Agence Nationale de la Recherche | ANR-12-BSV2-0005-02 | Rémi Dumollard<br>Céline Hebras<br>Lydia Besnardeau<br>Alex McDougall |
| Fondation ARC pour la Recherche sur le Cancer | SFI20111203776 | Rémi Dumollard<br>Céline Hebras<br>Lydia Besnardeau<br>Alex McDougall |

The funders had no role in study design, data collection and interpretation, or the decision to submit the work for publication.

### Author contributions

RD, Conceptualization, Data curation, Formal analysis, Funding acquisition, Investigation, Methodology, Project administration; NM, Formal analysis, Methodology, Developed the computational model; GS, SBA, Data curation, Formal analysis; FB, Data curation, Methodology; CH, LB, Methodology; AM, Conceptualization, Funding acquisition, Project administration

### Author ORCIDs

Rémi Dumollard, http://orcid.org/0000-0002-8444-0630

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
