## [Decision Letter]

Thank you for submitting your article "Cell cycle asynchrony determines cell shape at metaphase to support the invariant cleavage pattern of a chordate embryo" for consideration by *eLife*. Your article has been reviewed by three peer reviewers, and the evaluation has been overseen by Marianne Bronner as the Senior Editor. The reviewers have opted to remain anonymous.

The reviewers have discussed the reviews with one another and the Reviewing Editor has drafted this decision to help you prepare a revised submission.

Summary:

The authors seek to link cell shape, spindle orientation and timing of cell division to this, suggesting that these are connected and together constrain development. Cell adhesion is postulated as linking these and experiments manipulating adhesion via Ca^2+^ deprivation or playing with cell polarity support this, though the connection is indirect. The authors show that cleavage asynchrony is important for the shape of the embryo. They show this affects the shape of individual blastomeres. They contend that the orientation of cell division is controlled by cell shape, and show that when disrupted the development of the embryo (and hence setting up fated cell territories) goes awry.

While the experiments are persuasive and the flow of logic in the manuscript sound, there are key pieces of information missing. Buried in here is one assumption: that the shape of the embryo determines spindle orientation such that division occurs roughly along the longest axis. While data from cultured cells etc show this is a reasonable assumption, the data presented here only show a correlation between cell shape and spindle orientation. It is equally possible that both are determined by some other parameter (e.g. something similar happens in the ascidian germ line lineage where the CAB directs the division axis, as the authors consider and discuss).

Essential revisions:

1) The manuscript would be significantly strengthened if this connection could be experimentally demonstrated. For example, could they deform isolated blastomeres and/or embryos to 'stretch' cells in unusual directions? If the authors are correct this should also affect spindle orientation and the direction of cell division in a predictable way.

2) The Introduction is quite hard to follow. It should be expanded to provide important background issues/questions for the non-expert.

3) Subsection “Maternal and zygotic contributions to the stereotyped pattern of cell divisions.” – CAB depletion, removing cell-cell adhesion and arrest of zygotic transcription experiments: Do the cleavage patterns in these embryos obey the Hertwig's rule? I could not find clear statement about this point in the text. In the same section, the authors mentioned that the blastomeres exhibited misoriented cleavages. How are the cell division misoriented? I think the description here is somewhat vague and clearer explanation may be necessary. In addition, if the effect is particularly seen in cells that display spindle rotation in normal embryos (as stated in paragraph four of that section), there might be a mechanism other than Hertwig's rule to determine the orientation of spindle in these cells.

4) Subsection “Maternal and zygotic contributions to the stereotyped pattern of cell divisions.”, third paragraph: the significance of the experiments presented here. The authors mentioned that cell-cell interaction is vital for determining cleavage patterns are not clear. Additional results from the experiments such as Ca^2+^ free treatment may address this since embryos developed in Ca^2+^ free seawater are not able to keep their shape. What are the effect of the treatment on mitosis? Could positional changes of cells be caused by the lower resistance against the outside force due to the loss of cell-cell adhesion (for example, water flow could change cell position) while the cleavage patterns itself is conserved.

5) The authors claim that zygotic transcription is required for cell shaping and asynchrony of cell cycle up to the 64-cell stage by inhibiting zygotic transcription by suppressing β-catenin pathway and PEM overexpression. As both proteins are also involved in cellular functions other than controlling gene expression, this is not necessarily the case. Ideally, inhibition of zygotic transcription with pharmaceutical drugs, Actinomycin D or α-amanitin would be recommended. For example, in subsection “Maternal and zygotic contributions to the stereotyped pattern of cell divisions.”, fourth paragraph: the use of Pem1 for disrupting transcription of proteins may be problematic in this experiment, because Pem1 is a regulator of spindle position in Ciona (Negishi et al., 2007).

6) It is important to examine whether Wee1 is the target of TCF/β-catenin. This data could connect the β-catenin pathway, zygotic transcription, cell cycle asynchrony and cell division orientation. In the Discussion section, the authors mentioned that a gene regulatory network is co-opted to the long axis rule. However, the manuscript only states "transcription is necessary for the rule". This is an overstatement, since how the gene regulatory network is co-opted is not shown.

7) Subsection “Cell cycle asynchrony underpins the invariant spatial pattern of cell divisions” paragraph two: The authors argument is not clear, since mosaicism can be a significant cause of such variable phenotypes. Please discuss.

8) The authors should better explain the computational mathematical method they use to predict the spindle orientation. There is no information on this in the manuscript. At least, the basic logic should be provided in the present manuscript. The logic does not seem simple because in Figure 3, the predictions in the b4.2, A4.1, and B4.1 blastomeres do not seem to match the longest planer axis at a glance.

[Editors' note: further revisions were requested prior to acceptance, as described below.]

Thank you for resubmitting your work entitled "Cell cycle asynchrony determines cell shape at metaphase to support the invariant cleavage pattern of a chordate embryo" for further consideration at *eLife*. Your revised article has been favorably evaluated by Marianne Bronner (Senior editor), a Reviewing editor, and three reviewers. Because one of the original reviewers was not available for re-review, we sought the advice or a different reviewer who has raised some very good points. I have included both a summary of major points that need addressing, as well as the detailed comments of the reviewers (below) as that may be helpful to you in revising the manuscript.

The manuscript has been improved but there are some remaining issues that need to be addressed before acceptance, as outlined below:

Required Revisions:

1) The author's conclusion that cell division orientation follows the "long axis" rule is an oversimplification. It is more likely that cell division orientation is under a dual control: 1) apico-basal polarity specifies the orientation of the plane of division; 2) 2D cell geometry within this plane then constrains the orientation of cell division. This model departs from Hertwig's rule and appears quite different from the authors' proposal. Thus, the authors should remove the rigid statements that the ascidian's cleavage obeys the Hertwig's rule. The manuscript should be rewritten to indicate that cleavage orientation is determined by the shape (the longest axis) of the blastomeres along the observed plane.

2) It would be better to include Wee1 in situ hybridization data once again. While the authors used Wee1 MO and the dominant negative construct as the tools for modifying cell cycle, the analyses of the data need to be accompanied by the expression profile of the gene.

3) The Introduction is still very difficult to read and needs to be rewritten. Similarly, parts of the results need to better flow.

4) The title "Cell cycle asynchrony determines cell shape at metaphase" is not appropriate and needs to be changes, as the authors have not studied the impact on cell shape of changes in the timing of cell division.

Detailed comments from Reviewer #4:

Ascidian embryos develop with a stereotyped and invariant pattern of cell divisions. This study addresses the role of cellular geometry and timing in the orientation of these divisions. It combines experiments in early embryos with a modelling approach initially developed to characterize the plane of division of sea urchin eggs deformed into microfabricated chambers (Minc et al., 2011).

The study makes two major claims: 1) during normal development, the orientation of the division axis correlates with the long axis of the cell in its apical plane at metaphase, which is presented as corresponding to "Hertwig's rule" in the manuscript. This rule is respected when embryo morphology is altered by compression or ablation of a structure previously shown to orient divisions, the Contraction Pole. 2) The asynchrony of animal vs vegetal cell divisions is crucial for the precise oriented cell divisions, as observed by preventing the asynchrony through β-catenin or wee1 loss of function. In these experiments radical changes in both cell shape and planar division orientation are observed.

Overall, the paper is of lesser general interest, novelty and depth than ascidian articles recently published in *eLife* (e.g. Stolfi et al., 2014; Negishi et al., 2016, Hudson et al., 2016). The study identifies and quantifies interesting phenomena (some of which have previously been qualitatively described, e.g. cell rounding at mitosis, Tassy et al., 2006, Figure 4), but does not provide deep mechanistic insights. Authors present potentially interesting quantitative analyses of the evolution of mean cell shape during ascidians development, of spindle rotation at a specific stage, of mitotic cell rounding and apical expansion at metaphase in WT and perturbed embryos. The causal links between these different parameters are, however, not always clear and robust and the message thus lacks impact and depth. The two main messages, respect of Hertwig's rule and direct causal relationship between cell cycle timing, cell shape and orientation of cell divisions are insufficiently supported.

Major comments:

Hertwig's rule links the orientation of cell division to the 3D geometry of cells: 'The two poles of the division figure come to lie in the direction of the greatest protoplasmic mass' (Hertwig, 1884). This "long axis rule" is thus applicable to the 3D geometry of cells. Minc and colleagues (Cell, 2011) could use a 2D model of the process in Urchin eggs, because the height of the microfabricated wells used to constrain egg shape was smaller than the diameter of the egg, thereby making sure that the long axis of the cell was included in the analyzed plane. In the current study, the authors focus on planes perpendicular to the apico-basal axis of the cells, although the long axis of the cell may not be included in these planes. Figure 3 illustrates indeed that the long axis of most cells at metaphase is unlikely to lie in the plane of division (e.g. Figure 3 A6.3, a6.6). The author's conclusion that cell division orientation follows the "long axis" rule thus appears to be an oversimplification. It is more likely that cell division orientation is under a dual control: 1) apico-basal polarity specifies the orientation of the plane of division; 2) 2D cell geometry within this plane then constrains the orientation of cell division. This model departs from Hertwig's rule and appears quite different from the authors' proposal. Figure 2 may be consistent with the notion that when apico-basal polarity is affected, spindle orientation deviates from the expected plane (though the choice to perform the 2D analysis on the specific plane extracted in red appears quite arbitrary).

The planes chosen for geometric analysis are not always rigorously defined, and sometimes appear arbitrary. For example on Figure 1, the plane extracted for a6.8 is likely to be close to the apical plane, but those of a6.6 and b6.8 are at a significant angle from the apical planes of these cells. In Figure 5 control 8-cell, the extracted plane does not seem to follow the manual extraction rules presented in McDougall et al. (Methods Cell Biol 2015) as the apico-basal axis is included in – or close to – the selected plane. It is not clear how robust, reproducible and rigorous this manual method is. To avoid arbitrary choices, an automated method would be preferable. An analysis of the robustness of the model to changes in the definition of the plane would also help. It should also be said that the choice of the plane is not solely determined by the geometry of cell shape: the apico-basal position of the spindle is also taken into account.

The link between cell cycle asynchrony and the orientation of cell division is simply correlative and the impact of cell cycle asynchrony on cell shape control is not directly explored in this study, contrary to what the title suggests. The demonstration that affecting wee1 activity causes a speeding up of animal cell divisions and leads to changes in the orientation of cell divisions is interesting, but the analysis of these experiments remains too superficial. A likely explanation of the phenotype is that a change in the timing of cell divisions affects the shape of both animal and vegetal cells at metaphase and that this causes changes in their division orientation. The authors do not, however, analyze the shape of cells at metaphase following these perturbations, nor do they run their model to predict the new division axis.

Generally, the Introduction is difficult to read, as are some sections of the Results and Discussion. The question addressed at the end of the Introduction is not clearly defined. The text added to the Introduction during revision is a simple copy/paste from the article McDougall et al. (Meth Cell Biol. 2015) and is for the most part redundant. This is careless and unethical. The manuscript would benefit from a major rewriting effort. Figure layout could also be improved and is currently not homogeneous. It is difficult to understand the figures at first glance.

[Editors' note: further revisions were requested prior to acceptance, as described below.]

Thank you for submitting your article "Cell cycle asynchrony is required for robust spindle orientation in the planar axis at metaphase in a chordate embryo." for consideration by *eLife*. Your article has been reviewed by one peer reviewer, and the evaluation has been overseen by a Reviewing Editor and Marianne Bronner as the Senior Editor. The reviewers have opted to remain anonymous.

The Reviewing Editor has drafted this decision to help you prepare a revised submission.

Summary:

Unfortunately, we do not feel that you have adequately addressed the comments of the reviewers and that only minimal editorial changes were made. It is very unusual and strongly discouraged for a paper in *eLife* to go through this many rounds of review. Therefore, I will give you just one more chance to attend to the reviewers comments and otherwise will have to no choice but to decline your paper.

Essential revisions:

The authors have carried out a minimal textual revision of their work, involving minimal rewriting and have left several issues unanswered (e.g. would a different value for the threshold angle to call abnormal orientation qualitatively change the results?).

Issues that remain confusing and would need to be dealt with if the authors want to be understood by their readers are summarized below:

1) Planar axis: this is very confusing. Is it a plane? is it an axis (i.e. a line?). The definition the authors use is that of a plane, but they themselves get confused in their rebuttal letter. "We thank the reviewer for noting that it is not the cell's long axis but rather its longest planar axis…": here they do mean an axis (a plane has no length). But in the next paragraph "the cell's long axis occurs in the planar axis of cells…": here they mean a plane! I strongly suggest to entirely get rid of this confusing and unnecessary term and replace it by "apical plane" throughout the manuscript.

2) The authors still refer to "Hertwig's rule in the planar axis". They have also left ambiguous statements, about Hertwig's rule although the analysis is done in 2D. Hertwig's rule was defined as a 3D rule, and simply cannot be used in a 2D context. The authors should replace any mention to Hertwig's rule in a 2D context by a different term, saying for instance that the orientation of the cell division follows the long axis of the cell in the apical plane.

---

## [Author Response]

Essential revisions:

1) The manuscript would be significantly strengthened if this connection could be experimentally demonstrated. For example, could they deform isolated blastomeres and/or embryos to 'stretch' cells in unusual directions? If the authors are correct this should also affect spindle orientation and the direction of cell division in a predictable way.

This experiment has been performed and is now depicted in Figure 2. Embryos were compressed to cause unusual spindle orientations which were analysed with the computational model to show that these spindle orientation are oriented along the artificially created long axis of the cell.

*2) The Introduction is quite hard to follow. It should be expanded to provide important background issues/questions for the non-expert.*

The Introduction has been rewritten to provide more background.

3) Subsection “Maternal and zygotic contributions to the stereotyped pattern of cell divisions.” – CAB depletion, removing cell-cell adhesion and arrest of zygotic transcription experiments: Do the cleavage patterns in these embryos obey the Hertwig's rule? I could not find clear statement about this point in the text.

Cleavage patterns in CAB depleted and DA-aPKC expressing embryos have been analysed with the computational model to show that they still follow Hertwig’s rule (now depicted in Figure 5).

In the same section, the authors mentioned that the blastomeres exhibited misoriented cleavages. How are the cell division misoriented? I think the description here is somewhat vague and clearer explanation may be necessary.

Misoriented cleavages are cleavages that show different orientation from the invariant cleavage pattern consistently observed in control embryos of different ascidian species. This precision has been added in subsections “Maternal and zygotic contributions to the stereotyped pattern of cell divisions” and” Spindle position prediction using computational model (from Minc et al., 2011)” of the revised manuscript.

In addition, if the effect is particularly seen in cells that display spindle rotation in normal embryos (as stated in paragraph four of that section), there might be a mechanism other than Hertwig's rule to determine the orientation of spindle in these cells.

We are now showing computational analysis of misoriented cell divisions in DN-Tcf, PEM1(Figure 6) and Wee1KD (Figure 7) expressing embryos to show that cell shape is modified in these embryos and that misoriented cell divisions follow the Hertwig’s rule. Together with the observation that spindle rotates to align in the long planar axis of the cell (Figure 1) and that artificially changing cell shape can reorient spindles in a predictable way (Figure 2), this data suggests that stereotyped cell shape at metaphase drives spindle rotation during OCD. The higher incidence of misoriented cell divisions in lineages displaying OCD suggests to us that such stereotyped cell shape at metaphase (and hence Hertwig’s rule) is a major factor driving spindle rotation. The Discussion of the manuscript has been rewritten to emphasize this aspect.

4) Subsection “Maternal and zygotic contributions to the stereotyped pattern of cell divisions.”, third paragraph: the significance of the experiments presented here. The authors mentioned that cell-cell interaction is vital for determining cleavage patterns are not clear. Additional results from the experiments such as Ca^2+^ free treatment may address this since embryos developed in Ca^2+^ free seawater are not able to keep their shape. What are the effect of the treatment on mitosis? Could positional changes of cells be caused by the lower resistance against the outside force due to the loss of cell-cell adhesion (for example, water flow could change cell position) while the cleavage patterns itself is conserved.

Culture of ascidian embryos in Ca^2+^ free sea water does not affect the length of mitosis nor chromosome segregation suggesting that mitosis is not affected by decreased cell adhesion.

We now show in Figure 2 that 4 cell stage embryos with reduced cell adhesion (ie Ca free SW and DA-aPKC) display a similar topology (one layer of 4 cells) even though blastomeres are much rounder when cell adhesion is downregulated. In contrast the topology of 8 cell stage embryos is very different between control (2 layers of 4 cells) and adhesion depleted embryos. We analysed DA-aPKC embryos with the computational model for cell divisions occurring in the plane of imaging (ie 2 spindle poles are seen in the same z-plane) and found that the cell division followed the Hertwig’s rule. These observations suggest that the invariant cleavage pattern from 4 to 8 cell stage (which generates 2 layers of 4 cells) is supported by shape driven cell division and that cell adhesion is very important to implement the stereotypical cell shape at metaphase of mitosis from 4 to 8 cells.

*5) The authors claim that zygotic transcription is required for cell shaping and asynchrony of cell cycle up to the 64-cell stage by inhibiting zygotic transcription by suppressing β-catenin pathway and PEM overexpression. As both proteins are also involved in cellular functions other than controlling gene expression, this is not necessarily the case. Ideally, inhibition of zygotic transcription with pharmaceutical drugs, Actinomycin D or α-amanitin would be recommended. For example, in subsection “Maternal and zygotic contributions to the stereotyped pattern of cell divisions.”, fourth paragraph: the use of Pem1 for disrupting transcription of proteins may be problematic in this experiment, because Pem1 is a regulator of spindle position in Ciona (Negishi et al., 2007).*

We have used actinomycin D incubations and α-amanitin injections in Phallusia embryos but both drugs induced DNA bridges and variable slowing down of cell cycle (probably be due to S phase checkpoint activation) thus precluding the use of these drugs for our experiments. Such aspect is discussed in Dumollard et al., 2013 and we did not think necessary to mention this point again in this manuscript. We have now added a sentence referring to the 3 publications addressing this issue (shiriae-wrimayake et al., 2011, Kumano et al., 2011 and Dumollard et al., 2013)

We chose to use DN-Tcf to block only the transcriptional function of nuclear β-catenin without affecting cytosolic β-catenin. As the referee mention β-catenin morpholino (that we published in Dumollard et al., 2013) which affects both nuclear and cytosolic functions of β-catenin, changes cell cycle timing and downregulates cell adhesion. We therefore did not use β-catenin morpholino in this study.

Finally, it has been published by ourselves (Prodon et al., 2010; Dumollard et al., 2013) and by others (Negishi et al., 2007, Shirae-Kurabayashi et al., 2011, Kumano et al., 2011) that, even though depletion of PEM1 disrupts CAB-mediated unequal cleavage, PEM1 overexpression does not alter CAB activity resulting in 16 cell stage embryos with identical topologies between control and PEM1 expressing embryos (now mentioned in subsection “Maternal and zygotic contributions to the stereotyped pattern of cell divisions”). PEM1 expression did not induce DNA bridges and did not slow down cell cycle in the whole embryo suggesting that cell cycle is not impaired in these embryos and we think that PEM1 expression was the best tool we could use in *Phallusia* embryos to inhibit zygotic transcription.

6) It is important to examine whether Wee1 is the target of TCF/β-catenin. This data could connect the β-catenin pathway, zygotic transcription, cell cycle asynchrony and cell division orientation. In the Discussion, the authors mentioned that a gene regulatory network is co-opted to the long axis rule. However, the manuscript only states "transcription is necessary for the rule". This is an overstatement, since how the gene regulatory network is co-opted is not shown.

This sentence has been removed. We have also decided to remove the data concerning Wee1 and Cdc25 expression and the effect of Wee1-KD and Wee1 MO on gastrulation movements as suggested by reviewer #2 as they were not directly pertinent to the main finding of this study.

7) Subsection “Cell cycle asynchrony underpins the invariant spatial pattern of cell divisions” paragraph two: The authors argument is not clear, since mosaicism can be a significant cause of such variable phenotypes. Please discuss.

As the referee mentions cell shape in each blastomere might be affected differently by synchronization of the cell cycle which is a potential source of mosaicism. We have removed this point from the Discussion.

*8) The authors should better explain the computational mathematical method they use to predict the spindle orientation. There is no information on this in the manuscript. At least, the basic logic should be provided in the present manuscript. The logic does not seem simple because in Figure 3, the predictions in the b4.2, A4.1, and B4.1 blastomeres do not seem to match the longest planer axis at a glance.*

We have now added explanations of the designs underlying this model in the main text and in the material and methods. The model can integrate the effect of a complex geometry and, indeed, as observed by the referee the predictions may not systematically correspond to the longest axis in the cell, especially when this axis is not clearly defined (See Minc et al. *Cell* 2011).

[Editors' note: further revisions were requested prior to acceptance, as described below.]

Required Revisions:

1) The author's conclusion that cell division orientation follows the "long axis" rule is an oversimplification. It is more likely that cell division orientation is under a dual control: 1) apico-basal polarity specifies the orientation of the plane of division; 2) 2D cell geometry within this plane then constrains the orientation of cell division. This model departs from Hertwig's rule and appears quite different from the authors' proposal. Thus, the authors should remove the rigid statements that the ascidian's cleavage obeys the Hertwig's rule. The manuscript should be rewritten to indicate that cleavage orientation is determined by the shape (the longest axis) of the blastomeres along the observed plane.

We thank the reviewer for noting that it is not the cell’s long axis but rather its longest planar axis that directs spindle orientation. We agree and have changed the text to reflect this important point. Several vague statements like “regulation of cell division by cell shape” in our manuscript were confusing. We have now removed all confusing statements of the manuscript.

We now clearly emphasize that spindle orientation in the cell’s long axis occurs in the planar axis of cells which is supported by our data: “First, we found that every mitotic spindle tend to align with the long length of the cell’s planar axis at metaphase. Second, abolishing the asynchrony that causes the appearance of the 24-cell stage altered the invariant cleavage pattern. Third, misoriented cell divisions in synchronized embryos still followed the Hertwig’s rule in the planar axis.”).

Discussion on the mention of the Hertwig’s rule follows below.

Finally we thank the new reviewer for such a thorough review and for pointing out critical points in data analysis and interpretation. Responding to his/her remarks allowed us to improve seriously our manuscript.

2) It would be better to include Wee1 in situ hybridization data once again. While the authors used Wee1 MO and the dominant negative construct as the tools for modifying cell cycle, the analyses of the data need to be accompanied by the expression profile of the gene.

Pm-Wee1 in situ hybridization data is now included in Figure 7.

3) The Introduction is still very difficult to read and needs to be rewritten. Similarly, parts of the results need to better flow.

The Introduction clearly lacked a mention of regulation of spindle orientation by cell shape during planar cell divisions as performed in Campinho et al., 2013 (mitotic cell shape) or Bosveld et al., 2016 (interphasic cell shape), which used computational models modified from the one we use in this study.

Introduction length has been increased and Results section shortened and we are hoping now that important points are clear and introduced while we have removed details less pertinent to our main findings.

4) The title "Cell cycle asynchrony determines cell shape at metaphase" is not appropriate and needs to be changes, as the authors have not studied the impact on cell shape of changes in the timing of cell division.

We did assess the impact of cell cycle asynchrony on the shape of the apical surface of cells (Figure 8) but we did not study this matter systematically in every cells of the embryo so we changed the title to: “Cell cycle asynchrony is required for robust spindle orientation in the planar axis at metaphase in a chordate embryo.”

Detailed comments from Reviewer #4:

Ascidian embryos develop with a stereotyped and invariant pattern of cell divisions. This study addresses the role of cellular geometry and timing in the orientation of these divisions. It combines experiments in early embryos with a modelling approach initially developed to characterize the plane of division of sea urchin eggs deformed into microfabricated chambers (Minc et al., 2011).

The study makes two major claims: 1) during normal development, the orientation of the division axis correlates with the long axis of the cell in its apical plane at metaphase, which is presented as corresponding to "Hertwig's rule" in the manuscript. This rule is respected when embryo morphology is altered by compression or ablation of a structure previously shown to orient divisions, the Contraction Pole. 2) The asynchrony of animal vs vegetal cell divisions is crucial for the precise oriented cell divisions, as observed by preventing the asynchrony through β-catenin or wee1 loss of function. In these experiments radical changes in both cell shape and planar division orientation are observed.

Overall, the paper is of lesser general interest, novelty and depth than ascidian articles recently published in eLife (e.g. Stolfi et al., 2014; Negishi et al., 2016, Hudson et al., 2016). The study identifies and quantifies interesting phenomena (some of which have previously been qualitatively described, e.g. cell rounding at mitosis, Tassy et al., 2006, Figure 4), but does not provide deep mechanistic insights. Authors present potentially interesting quantitative analyses of the evolution of mean cell shape during ascidians development, of spindle rotation at a specific stage, of mitotic cell rounding and apical expansion at metaphase in WT and perturbed embryos. The causal links between these different parameters are, however, not always clear and robust and the message thus lacks impact and depth. The two main messages, respect of Hertwig's rule and direct causal relationship between cell cycle timing, cell shape and orientation of cell divisions are insufficiently supported.

We thank the reviewer for this suggestion on the Tassy et al. 2006 study on another ascidian species (*Ciona intestinalis*) which we clearly omitted to mention in our manuscript. Tassy et al. indeed shows changes in cell shape at the 32-44 cell stages, Figure 4 in particular shows cell shape changes for A6.1 and its animal opposing cell a6.8 which are the same as we have observed in *Phallusia* embryos. In Tassy et al. the authors note “Amusingly, between the early 32-cell stage and the mid 44-cell stage, the elongation factors of opposing animal and vegetal cells change in precisely opposite manner with time as exemplified for a6.8 and A6.1…”. We find in our study that such mirror behaviors between animal and vegetal blastomeres is brought about by cell cycle asynchrony. By analyzing these cell shape changes in relation to the cell cycle and to apico-basal polarity our study reveals that these cell shape changes are actually related to mitotic cell rounding and apical expansion (as observed in *Drosophila* epithelia) (Figure 3). Our study also suggests that the mirror changes in elongation of opposing An/Veg blastomeres noted by Tassy et al. are crucial for morphogenesis of the ascidian blastula as removing them (by synchronizing the embryo) impairs the invariant cleavage pattern.

We have added the following sentence to the manuscript:

Introduction section: “Major cell shape changes have been noted during the 32-44-cell stage in ascidian (*Ciona*) embryos (Tassy et al., 2006). Such cell shape changes may be related to mitotic cell rounding and may impact the highly predictable pattern of oriented cell divisions (OCDs) generating the 64 cell ascidian blastula especially in the context of asynchronous cell divisions observed at the 16-32-44-cell stages.”

Discussion section: “Cell shape changes in 32-44-cell stage ascidian embryos were noted previously (Tassy et al., 2006), prompting us to test the idea that cell cycle stage may be involved in their mirror-like cell shape regulation”

The aspects of mitotic cell rounding and cell cycle asynchrony were not addressed in Tassy et al. study and we believe the main findings of our study are completely novel.

Major comments:

Hertwig's rule links the orientation of cell division to the 3D geometry of cells: 'The two poles of the division figure come to lie in the direction of the greatest protoplasmic mass' (Hertwig, 1884). This "long axis rule" is thus applicable to the 3D geometry of cells. Minc and colleagues (Cell, 2011) could use a 2D model of the process in Urchin eggs, because the height of the microfabricated wells used to constrain egg shape was smaller than the diameter of the egg, thereby making sure that the long axis of the cell was included in the analyzed plane. In the current study, the authors focus on planes perpendicular to the apico-basal axis of the cells, although the long axis of the cell may not be included in these planes. Figure 3 illustrates indeed that the long axis of most cells at metaphase is unlikely to lie in the plane of division (e.g. Figure 3 A6.3, a6.6). The author's conclusion that cell division orientation follows the "long axis" rule thus appears to be an oversimplification.

The reviewer is correct and we have removed all ambiguity in the language used to convey the idea that the spindle lies within the cell’s long axis. Our data show that the spindle lies within the cell’s long length in the planar axis (or apical plane). The reviewer is right to note that Figure 3 suggests that the long axis of most cells at metaphase is unlikely to lie in the plane of division (ie the planar axis or the apical plane) as only one view of 4 blastomeres were shown. We have actually extracted the apico-basal axis of each blastomere and analyzed them with the computational model and we found that the long axis correlates within 30° with the planar axis (or apical plane) in all cells at the 2, 4, 8, 16c and 32c vegetal stages. In the ectoderm (animal, the 44 cell) all cells but 3 showed an alignment with the long axis of the cell. Because of these 3 exceptions we are now investigating the role of apicobasal polarity, junctional complexes and acto-myosin cortex in the planar orientation of spindle. This is the subject of our ongoing study.

It is more likely that cell division orientation is under a dual control: 1) apico-basal polarity specifies the orientation of the plane of division; 2) 2D cell geometry within this plane then constrains the orientation of cell division. This model departs from Hertwig's rule and appears quite different from the authors' proposal.

Without additional data in the present manuscript the hypotheses of the reviewer are the most parsimonious. However we prefer to not comment on the aspect of planar spindle positioning in the present manuscript (see previous comment).

The Hertwig’s rule (also coined “long axis rule”) has been tested on 2D samples and 2D images (Hertwig, 1884; Strauss et al., 2006; Minc et al., 2011; Campinho et al., 2013) and we have now explained that, in our manuscript, the long axis rule applies only for spindle rotation/orientation in the planar axis of cells as it applies for spindle orientation in the planar axis of zebrafish epibolizing EVL cells (Campinho et al., 2013).

We have nevertheless rewritten the whole manuscript to remove the ambiguity between 3D cell shape and shape of the 2D apical plane (or planar axis) as it was indeed confusing.

Figure 2 may be consistent with the notion that when apico-basal polarity is affected, spindle orientation deviates from the expected plane (though the choice to perform the 2D analysis on the specific plane extracted in red appears quite arbitrary).

Figure 2 shows that when basolateral membrane formation is inhibited cell adhesion is reduced at the 4 cell stage and cells come close to a round shape. Nevertheless when cell division occurred in the imaging plane we could analyze them using 2D imaging and the 2D computational model and found a good correlation between the orientations of observed and predicted spindles.

The planes chosen for geometric analysis are not always rigorously defined, and sometimes appear arbitrary. For example on Figure 1, the plane extracted for a6.8 is likely to be close to the apical plane, but those of a6.6 and b6.8 are at a significant angle from the apical planes of these cells. In Figure 5 control 8-cell, the extracted plane does not seem to follow the manual extraction rules presented in McDougall et al. (Methods Cell Biol 2015) as the apico-basal axis is included in – or close to – the selected plane. It is not clear how robust, reproducible and rigorous this manual method is. To avoid arbitrary choices, an automated method would be preferable. An analysis of the robustness of the model to changes in the definition of the plane would also help. It should also be said that the choice of the plane is not solely determined by the geometry of cell shape: the apico-basal position of the spindle is also taken into account.

Figure 1, Figure 2, Figure 5, Figure 6 and Figure 8 depict 2D imaging experiments and only the cells whose spindle remains in the imaging plane during the whole of mitosis were analysed. The manual extraction method we published in McDougall et al. (Methods Cell Biol 2015) was used only for Figure 3 and 4. We agree with the reviewer that the use of both 2D and 3D imaging experiments throughout the manuscript is confusing. This reflects the rather low tech/cost study we have performed using manual segmentation, manual 3D rendering and manual computational analysis which precludes a more exhaustive testing of the protocols (as afforded by high tech automated analysis). For the same reason (time/cost effectiveness) we have used 2D imaging experiments when a systematic 3D imaging experiment/analysis would have been more informative.

During the course of this work we have compared 2D and 3D analysis and found that for blastomeres at the 8 cell stage and for blastomeres whose apical plane is close enough to the imaging plane (ie. A5.1, B5.1, a5.4, b5.4, A6.1, B6.1, a6.8, b6.8 and often a6.6) centering and orienting deviations values were similar between 2D and 3D imaging experiments. In contrast for other blastomeres (medial ones: A5.2, B5.2, a5.3, b5.3, A6.3, A6.4, B6.2, B6.3, B6.4, a6.5, a6.7, b6.6 and b6.7), extraction of the planar axis during 3D imaging experiments resulted in much lower orienting deviations than in 2D experiments.

Therefore there is a bias in our study towards the blastomeres whose apical surface and spindles were contained in the imaging plane during 2D imaging experiments and we undertook 3D analysis to extend our study to medial blastomeres. With our lengthy manual 3D analysis we could not have analysed all the embryos used for the whole study. We are now implementing the tools necessary for 3D live imaging and automated 3D analysis in order to circumvent this problem in the future.

The link between cell cycle asynchrony and the orientation of cell division is simply correlative and the impact of cell cycle asynchrony on cell shape control is not directly explored in this study, contrary to what the title suggests. The demonstration that affecting wee1 activity causes a speeding up of animal cell divisions and leads to changes in the orientation of cell divisions is interesting, but the analysis of these experiments remains too superficial. A likely explanation of the phenotype is that a change in the timing of cell divisions affects the shape of both animal and vegetal cells at metaphase and that this causes changes in their division orientation. The authors do not, however, analyze the shape of cells at metaphase following these perturbations, nor do they run their model to predict the new division axis.

Our observations show that: “First, we found that every mitotic spindle tends to align with the long length of the cell’s planar axis at metaphase. Second, abolishing the asynchrony that causes the appearance of the 24-cell stage altered the invariant cleavage pattern. Third, misoriented cell divisions in synchronized embryos still followed the Hertwig’s rule in the planar axis.”.

We would therefore like to emphasize that we did analyze the shape of cells at metaphase following synchronisation, and we did run the computational model to predict the new division axis but only on a6.8.

We agree with the reviewer that a more exhaustive study would be welcome. In light of the reviewers comments, we therefore changed the title of our manuscript to remove the mention of cell shape.

Generally, the Introduction is difficult to read, as are some sections of the Results and Discussion. The question addressed at the end of the Introduction is not clearly defined. The text added to the Introduction during revision is a simple copy/paste from the article McDougall et al. (Meth Cell Biol. 2015) and is for the most part redundant. This is careless and unethical. The manuscript would benefit from a major rewriting effort. Figure layout could also be improved and is currently not homogeneous. It is difficult to understand the figures at first glance.

Sentence changed to: “Even after such genomic divergence, ascidian embryos from different species have a conserved invariant cleavage pattern and the system for naming the different blastomeres up to the gastrula stage developed by Conklin in 1905 for a stolidobranch ascidian (*Styela partita*, Conklin, 1905) is currently used for phlebobranch embryos (*Ciona, Phallusia*)”.

[Editors' note: further revisions were requested prior to acceptance, as described below.]

Essential revisions:

The authors have carried out a minimal textual revision of their work, involving minimal rewriting and have left several issues unanswered (e.g. would a different value for the threshold angle to call abnormal orientation qualitatively change the results?).

We thank the reviewer for this comment. We show an analysis of the data which now appears in a revised Figure 4. Here in Figure 4 149 cell divisions from the 2 to the 44-cell stage are represented (not including the unequal cell divisions in B4.1/B5.2/B6.3). This graph shows that 88% of analysed cell divisions are accurately predicted to be regulated by cell shape with a threshold of 30°. If setting a threshold of 20°, 78% of equal cell divisions are predicted to be regulated by cell shape. Even with a 10° threshold for than 50% of all equal cell divisions were accurately predicted by our model. We also performed a Kolgomorov-Smirnov test to show that the distribution of orienting deviations found in these 149 equal cell divisions are significantly different from a random distribution of angles.

We have added a paragraph in the Result section:

“By comparing the orienting deviation observed in 149 cell divisions (excluding the germ cell precursors that divide unequally) to randomly generated angles between 0 and 90° we found that the distribution of observed orienting deviations is non uniform and significantly different from a random distribution (Figure 4). In our data set, 88% of cells show an orienting deviation of 30° or less, 78% of cells have an orienting deviation under 20° and 52% of cells show an orienting deviation of less than 10° (Figure 4). Therefore our model robustly predicts that spindle aligns with the long length of the apical surface in most cells with a precision of 30° and 20°. In contrast spindle orientation could not be predicted reliably in B5.2, B6.3 and A6.3 blastomeres (i. e. orienting deviation is above 30°) suggesting that in these three blastomeres apical cell shape does not regulate spindle positioning in the apical plane.”

And in the Discussion section.

“After confirming in ascidian embryos that, like in *C. elegans*, sea urchin, Zebrafish or *Xenopus* embryos (Wildwater et al., 2011; Minc et al., 2011, Strauss et al., 2006, Pierre et al., 2016), cell shape impacts the orientation of cell division, we show that a computational model based on the shape of the apical surface predicts spindle orientation in the apical surface in all cells of the embryo except in the germ lineage (B5.2, B6.3) and A6.3. 88% of these cells can be predicted by the model with a 30° precision and 78% of these cells are still predicted by the computational model with 20° precision (Figure 4). These observations strongly indicate that mitotic spindles align with the long axis of the cell in the apical plane to implement the invariant cleavage pattern of planar cell division thus creating the distinctive topographical organization of cells in the ascidian blastula.”

Issues that remain confusing and would need to be dealt with if the authors want to be understood by their readers are summarized below:

1) Planar axis: this is very confusing. Is it a plane? is it an axis (i.e. a line?). The definition the authors use is that of a plane, but they themselves get confused in their rebuttal letter. "We thank the reviewer for noting that it is not the cell's long axis but rather its longest planar axis…": here they do mean an axis (a plane has no length). But in the next paragraph "the cell's long axis occurs in the planar axis of cells…": here they mean a plane! I strongly suggest to entirely get rid of this confusing and unnecessary term and replace it by "apical plane" throughout the manuscript.

We agree that “…the planar axis…” was the wrong term for the reasons mentioned by the reviewer. We have now replaced all statements such as “long planar axis” by “long length of the cell in the apical plane” throughout the manuscript. “planar axis” was replaced by “apical plane” or “apical surface”.

2) The authors still refer to "Hertwig's rule in the planar axis". They have also left ambiguous statements, about Hertwig's rule although the analysis is done in 2D. Hertwig's rule was defined as a 3D rule, and simply cannot be used in a 2D context. The authors should replace any mention to Hertwig's rule in a 2D context by a different term, saying for instance that the orientation of the cell division follows the long axis of the cell in the apical plane.

We agree again with the reviewer and have removed all references to the Hertwig’s rule as it pertains to the data presented from the manuscript and changed to “follows the long axis of the cell in the apical plane” or by “spindle positioning is regulated by apical cell shape”.

We have also rewritten a large part of the Introduction section which should be more relevant to our data and didactical for non expert readers.